# A Halfspace-Mass Depth-Based Method for Adversarial Attack Detection

**Marine Picot**[*]                                             *marine.picot@{centralesupelec.fr, mail.mcgill.ca}*
*Laboratoire des Signaux et Systèmes (L2S), Université Paris-Saclay CNRS CentraleSupélec*
*Department of Electrical and Computer Science McGill University, QC, Canada*

**Federica Granese**[*]                                                         *federica.granese@inria.fr*
*Lix, Inria, Institute Polytechnique de Paris, Sapienza University of Rome*

**Guillaume Staerman**                                                      *guillaume.staerman@inria.fr*
*Université Paris-Saclay, Inria, CEA*

**Marco Romanelli**                                                              *mr6852@nyu.edu*
*Laboratoire des Signaux et Systèmes (L2S), Université Paris-Saclay CNRS CentraleSupélec*

**Francisco Messina**                                                             *fmessina@fi.uba.ar*
*School of Engineering, Universidad de Buenos Aires*
*CSC-CONICET, Buenos Aires, Argentina*

**Pablo Piantanida**                                                         *pablo.piantanida@cnrs.fr*
*International Laboratory on Learning Systems (ILLS), CNRS, CentraleSupélec*

**Pierre Colombo**                                                     *pierre.colombo@centralesupelec.fr*
*MICS, CentraleSupélec*

**Reviewed on OpenReview:** *https://openreview.net/forum?id=YtUOnDb5e8*

## Abstract

Despite the widespread use of deep learning algorithms, vulnerability to adversarial attacks is still an issue limiting their use in critical applications. Detecting these attacks is thus crucial to build reliable algorithms and has received increasing attention in the last few years. In this paper, we introduce the **H**alfsp**A**ce **M**ass de**P**th d**E**tecto**R** (`HAMPER`), a new method to detect adversarial examples by leveraging the concept of data depths, a statistical notion that provides center-outward ordering of points with respect to (w.r.t.) a probability distribution. In particular, the halfspace-mass (HM) depth exhibits attractive properties which makes it a natural candidate for adversarial attack detection in high-dimensional spaces. Additionally, HM is non differentiable making it harder for attackers to directly attack `HAMPER` via gradient based-methods. We evaluate `HAMPER` in the context of supervised adversarial attacks detection across four benchmark datasets. Overall, we empirically show that `HAMPER` consistently outperforms SOTA methods. In particular, the gains are 13.1% (29.0%) in terms of AUROC↑ (resp. FPR $\downarrow_{95\%}$) on SVHN, 14.6% (25.7%) on CIFAR10 and 22.6% (49.0%) on CIFAR100 compared to the best performing method.

## 1 Introduction

In most machine learning applications, deep models have achieved state-of-the-art performance. However, an important limitation to their widespread use in critical systems is their vulnerability to adversarial attacks (Szegedy et al., 2014), i.e., the introduction of maliciously designed data crafted through minor adversarial perturbations to deceive a trained model. This phenomenon may lead to disastrous consequences

in sensitive applications such as autonomous driving, aviation safety management, or health monitoring systems (Geifman & El-Yaniv, 2019; Geifman et al., 2019; Guo et al., 2017; Meinke & Hein, 2020).

Over time, a vast literature has been produced on defense methods against adversarial examples (Croce & Hein, 2020; Athalye et al., 2018b; Zheng et al., 2019; Aldahdooh et al., 2021b). On the one hand, techniques to train models with improved robustness to upcoming attacks have been proposed in Zheng et al. (2016); Madry et al. (2018) or Picot et al. (2021). On the other hand, effective methods to detect adversarial examples given a pre-trained model were reported in Kherchouche et al. (2020); Meng & Chen (2017) or Ma et al. (2019). Detection methods for adversarial examples can be mainly grouped into two categories (Aldahdooh et al., 2021b): *supervised* and *unsupervised* ones. In the *supervised* detection setting, the detector is trained on features extracted from adversarial examples generated according to one or multiple attacks. In particular, the *network invariant model approach* consists of features that are derived from the activation values of the network layers (cf. Lu et al., 2017; Carrara et al., 2018 or Metzen et al., 2017); in the *statistical approach* the features are linked to in-training or out-of-training data distribution/manifold (e.g., maximum mean discrepancy (Grosse et al., 2017), PCA (Li & Li, 2017), kernel density estimation (Feinman et al., 2017), local intrinsic dimensionality (Ma et al., 2018), latent graph neighbors (Abusnaina et al., 2021), the Mahanalobis distance (Lee et al., 2018); in the *auxiliary model approach*, the features are instead

derived from monitoring clean and adversarial characteristics (e.g., model uncertainty (Feinman et al., 2017), natural scene statistics (Kherchouche et al., 2020)). In the *unsupervised* detection setting, the detector does not rely on the prior knowledge of the attacks, and it only learns from the clean data at training time. Different techniques are used to extract the meaningful features (e.g., *feature squeezing* (Xu et al., 2018; Liang et al., 2021), *denoiser approach* (Meng & Chen, 2017), *network invariant* (Ma et al., 2019), *sensitivity to noise* (Hu et al., 2019), *auxiliary model* (Sotgiu et al., 2020; Aldahdooh et al., 2021a; Zheng & Hong, 2018), *k-Nearest Neighbors* (Raghuram et al., 2021)). Detection methods of adversarial examples differ as well according to whether the underlying classifier is assumed to be pre-trained or not: when further training of the classifier is allowed, the methods present *a novel training procedure* (e.g., with reverse cross-entropy (Pang et al., 2018); with the rejection option (Sotgiu et al., 2020; Aldahdooh et al., 2021a)) and a thresholding test strategy. Finally, the learning task of the underlying network also impacts the adversarial examples detection methods (e.g., detection of adversarial examples for human recognition tasks (Tao et al., 2018)). Adversarial detection can be related to the *anomaly detection* problem. Indeed, anomaly detection aims to identify abnormal observations without previously knowing them, possibly including adversarial attacks. A plethora of techniques has been designed to address this problem ranging from machine learning algorithms

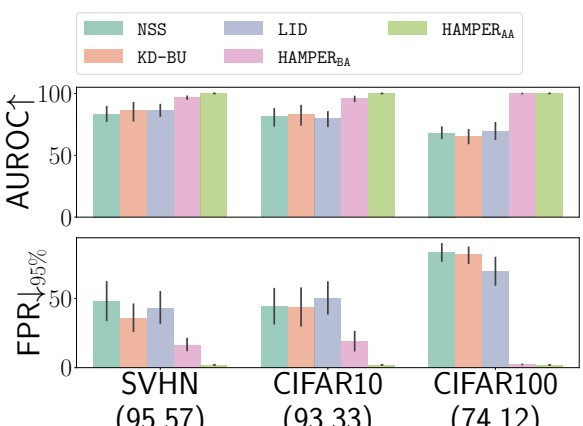

Figure 1: Average performances of our method (i.e., `HAMPER`) in an attack-aware and single detector setting, along with the performances of state-of-the-art detection mechanisms (i.e., `NSS`, `LID`, `KD-BU`), on three classically considered datasets (i.e., SVHN, CIFAR10 and CIFAR100). Below the dataset names is the accuracy of their underlying classifiers. In addition to outperforming other methods on all three considered datasets, our method, contrary to the others, does not lose performances as the classifier's accuracy decreases.

such as Isolation Forest (Liu et al., 2008; Staerman et al., 2019), Local Outlier Factor (Breunig et al., 2000) or One-Class SVM (Schölkopf et al., 2001) to statistical tools such as kernel density estimation (Feinman et al., 2017) or Data Depth (Zuo & Serfling, 2000) (see Chandola et al., 2009 for an extensive review of anomaly detection methods). In particular, data depth stands out as a natural candidate to detect anomalies (Chen et al., 2009).

The idea of statistical depth has grown in popularity in multivariate data analysis since its introduction by John Tukey (Tukey, 1975). For a distribution on $\mathbb{R}^d$ with $d > 1$, by transporting the natural order on the real

line to $\mathbb{R}^d$, a depth function provides a center-outward ordering of points w.r.t. the distribution. *The higher the point depth score, the deeper the point is in the distribution.* In addition to anomaly detection (Serfling, 2006; Staerman et al., 2020; Chen et al., 2009; Staerman et al., 2021b; 2022), the notion of depth has been used to extend the notions of (signed) rank or order statistics to multivariate data, which find numerous applications in statistics and machine learning (e.g. robust inference (Cuevas et al., 2007), classification (Lange et al., 2014), hypothesis testing (Oja, 1983), clustering (Jörnsten, 2004; Staerman et al., 2021a). To the best of our knowledge, it has not been investigated yet through the lens of adversarial attack detection. This paper aims to leverage this overlooked notion to build an adversarial attack detector.

**Contributions.** Our contribution is threefold:

1. We propose applying the halfspace-mass depth notion in the context of the adversarial detection problem. To the best of our knowledge, we are the first to both explore and successfully apply data depth for adversarial detection.

2. Through an analysis of the classifier's behavior under threat, we show how to leverage the halfspace-mass depth to build an anomaly score. To that end, we introduce `HAMPER`, a simple supervised method to detect adversarial examples given a trained model. Given an input sample, `HAMPER` relies on a linear combination of the halfspace-mass depth score. These depth scores are computed w.r.t. a reference distribution corresponding to the training data conditioned per-class and per-layer.

3. We extensively evaluate `HAMPER`'s performance across popular attack strategies and computer vision benchmark datasets (e.g., SVHN, CIFAR10, and CIFAR100). As shown by Fig. 1, `HAMPER` largely outperforms SOTA detection methods and consistently detects attacks that SOTA approaches fail to identify.

The paper is organized as follows. First, in Sec. 2, we describe the adversarial detection problem and provide a detailed overview of the SOTA supervised detection methods and the attack mechanisms considered throughout the paper. In Sec. 3, after recalling the concept of data depth by focusing on the halfspace-mass depth, we introduce `HAMPER`, our proposed supervised detector method based on the HM depth. In Sec. 4, we provide insights on the underlying classifier's behavior under threats. In Sec. 5, we extensively evaluate `HAMPER` through numerical experiments on benchmarks on visual datasets and compare it to SOTA methods. Finally, concluding remarks are gathered in Sec. 6.

## 2 Background

After defining the problem formulation, we present the SOTA detection methods and the attack mechanisms that we will consider throughout this paper.

### 2.1 Problem Formulation

Let $(X, Y)$ be a random tuple of variables valued in $\mathcal{X} \times \mathcal{Y}$ with unknown data distribution $p_{XY}$; $\mathcal{X} \subset \mathbb{R}^d$ represents the feature space and $\mathcal{Y} = \{1, \ldots, C\}$ represents the labels attached to elements in $\mathcal{X}$, where $C \in \mathbb{N}$, $C \geq 2$. The training dataset $\mathcal{D} = \{(x_i, y_i)\}_{i=1}^n$ is defined as $n \geq 1$ independent identically distributed (i.i.d.) realizations of $p_{XY}$. Subsets of the feature space associated with a label $c \in \mathcal{Y}$ are denoted by $\mathcal{S}_c = \{x_i \in \mathcal{S} : y_i = c\}$ with $\mathcal{S} = \{x_i\}_{i=1}^n$.

Given a parametric model with $L \geq 1$ layers; let $f_\theta^\ell : \mathcal{X} \to \mathbb{R}^{d_\ell}$ with $\ell \in \{1, \ldots, L\}$, denotes the output of the $\ell$-th layer of the deep neural network (DNN) parametrized by $\theta \in \Theta$ where the dimension of the latent space induced by the $\ell$-th layer is $d_\ell$. The class prediction is obtained from the $L$-th layer softmax output as follows:

$$f_\theta^L(x; \mathcal{D}) \triangleq \arg\max_{c \in \mathcal{Y}} q_\theta(c|x) \text{ with } q_\theta(\cdot|x) = \text{softmax}(f_\theta^{L-1}(x; \mathcal{D})).$$

**The adversarial problem.** Given $x \in \mathcal{X}$ and $p \geq 1$, the adversarial generation problem can be defined as producing $x'$ such as (Szegedy et al., 2014):

$$x' = \arg\min_{x' \in \mathbb{R}^d : \|x' - x\|_p < \varepsilon} \|x' - x\| \text{ s.t. } f_\theta^L(x'; \mathcal{D}) \neq y, \tag{1}$$

where $y$ is the true label associated to the sample $x$, and $\|\cdot\|_p$ is the $p$-norm operator. Since this problem is computationally infeasible in general, it is commonly relaxed as follows:

$$x' = \underset{x' \in \mathbb{R}^d \, : \, \|x'-x\|_p < \varepsilon}{\arg\max} \mathcal{L}(x, x'; \theta), \tag{2}$$

where $\mathcal{L}(x, x'; \theta)$ is the objective of the attacker, representing a surrogate of the constraint to fool the classifier, i.e., $f_\theta^L(x'; \mathcal{D}) \neq y$. The variety of attacks differs with the choice of the norm (e.g., $p = 1, 2, \infty$) and the value of $\varepsilon$.

**Crafting a detector.** Given a new observation $x \in \mathbb{R}^d$, detecting adversarial attacks boils down to build a binary rule $g : \mathbb{R}^d \to \{0, 1\}$. Namely: a new observation $x \in \mathbb{R}^d$ is considered as 'normal' (or 'natural', 'clean'), i.e. generated by $p_{XY}$, when $g(x) = a$ with $a \in \{0, 1\}$, and $x$ is considered as an adversarial example when $g(x) = 1 - a$. For a given scoring function $s : \mathbb{R}^d \to \mathbb{R}$, and a threshold $\gamma \in \mathbb{R}$, we have

$$g(x) = \mathbb{I}\{s(x) > \gamma\} = \begin{cases} 1 \text{ if } s(x) > \gamma, \\ 0 \text{ if } s(x) \leq \gamma. \end{cases} \tag{3}$$

## 2.2 Supervised Detection Methods

Supervised detection methods, when the defender has access to the future threats that it is going to face, can be separated into two main groups: attack-aware and blind-to-attack methods.

**Attack-aware methods.** In the attack-aware setting, the methods are going to face a single threat, and they have full knowledge about them. Therefore, it is possible to train one detector per attack. In this setting fall two detection methods: KD-BU (Feinman et al., 2017) and LID (Ma et al., 2018). KD-BU is based on the intuition that the adversarial examples lie off the data manifold. To train the detector, a *kernel density* estimation in the feature space of the last hidden layer is performed, followed by an estimation of the *bayesian uncertainty* of the input sample. LID extracts the *local intrinsic dimensionality* features for natural and attacked samples for each layer of the classifier and trains a detector on them.

**Blind-to-Attack setting.** In the blind-to-attack setting, the defender has knowledge about the fact that it is going to be attacked, but do not know exactly how. In that case, a single detector is trained, deployed and tested against all possible threats. NSS (Kherchouche et al., 2020) falls into that category. It is based on the extraction of the *natural scene statistics* from the clean and adversarial samples from different threats models, later used to train a detector to discriminate between natural inputs and adversarial examples. Natural scene statistics are regular statistical properties that are altered by adversarial perturbations.

## 2.3 A Brief Review of Attack Mechanisms

Multiple methods to generate adversarial examples have been developed in recent years. The attack mechanisms can be divided into two main categories: whitebox, where the attacker has complete knowledge about the targeted classifier, and blackbox, where the attacker does not know about the targeted classifier.

**Whitebox attacks.** The simplest one is Fast Gradient Sign Method (**FGSM**), introduced by Goodfellow *et al.* (Goodfellow et al., 2015). It consists in modifying the examples in the direction of the gradient of a specific objective, w.r.t. the input on the targeted classifier. Two iterative versions of FGSM have been proposed: Basic Iterative Method (**BIM**; Kurakin et al., 2018) and Projected Gradient Descent (**PGD**; Madry et al., 2018). The main difference is that BIM initializes the adversarial example to the natural sample while PGD initializes it to the natural example plus random noise. Although PGD was initially created under an $L_\infty$ constraint, it is possible to extend the method to any $L_p$-norm constraint. Later, Moosavi-Dezfooli *et al.* (Moosavi-Dezfooli et al., 2016) introduced DeepFool (**DF**), an iterative method based, at each step, on a local linearization of the model, resulting in a simplified problem. Finally, Carlini-Wagner (Carlini & Wagner, 2017) presents the **CW** method to find the smallest noise solving the original adversarial problem. They proposed a new relaxed version of the adversarial problem that optimizes an attack objective, chosen according to a specific task.

**Blackbox attacks.** Without any knowledge about the targeted classifier or its gradients, blackbox attacks are expected to rely on different mechanisms. Square Attack (**SA**; Andriushchenko et al., 2020) employs a random search for perturbations that maximize a given objective, Spatial Transformation Attack (**STA**; Engstrom et al., 2019) applies small translations and rotations to the original image while Hop Skip Jump (**HOP**; Chen et al., 2020) estimates the gradient-based direction to perturb through a query on the targeted classifier.

**Adaptive attacks.** There exists a third type of attacks called **Adaptive Attacks** (Athalye et al., 2018a; Tramer et al., 2020; Carlini & Wagner, 2017; Yao et al., 2021). Adaptive attacks have full knowledge about not only the underlying classifier to attack, but also the defense mechanisms one may have deployed. To build efficient adaptive attacks, it is therefore crucial to understand the mechanisms involved into the defense, and finding ways to bypass them. For examples, the Backward-Pass Differentiable Attack (**BPDA**; Athalye et al., 2018a) has been developed to overcome the non-differentiability of the defense mechanisms by finding a suitable surrogate to the non-differential parts of them.

## 3 A Depth-Based Detector

After presenting the data depth in Sec. 3.1, with an emphasis on the halfspace-mass depth, we introduce our depth-based detector in Sec. 3.2.

### 3.1 Background on Data-Depth

A data depth function $D(\cdot, P) : \mathbb{R}^d \to [0, 1]$ measures the centrality of any element in $x \in \mathbb{R}^d$ w.r.t. a probability distribution $P$ (respectively, a data set). It provides a center-outward ordering of points in the support of $P$ and can be straightforwardly used to extend the notions of rank or order statistics to multivariate data. The higher $D(x, P)$, the deeper $x \in \mathbb{R}^d$ is in $P$. The earliest proposal is the *halfspace* depth introduced by John Tukey in 1975 (Tukey, 1975). This depth is very popular due to its appealing properties and ease of interpretation. Assume that $P$ is defined on an arbitrary subset $\mathcal{K} \subset \mathbb{R}^d$ and denote by $P(H) \triangleq P(H \cap \mathcal{K})$ the probability mass of the closed halfspace $H$. The halfspace depth of a point $x \in \mathbb{R}^d$ with respect to a probability distribution $P$ on $\mathbb{R}^d$ is defined as the smallest probability mass that can be contained in a closed halfspace containing $x$:

$$D_{\mathrm{H}}(x, P) = \inf_{H \in \mathcal{H}(x)} P(H),$$

where $\mathcal{H}(x)$ is the set of all closed halfspaces containing $x$.

However, the halfspace depth suffers from three critical issues: ($i$) finding the direction achieving the minimum to assign it a score induces a significant sensitivity to noisy directions, ($ii$) assigning the zero score to each new data point located on the outside of the convex hull of the support of $P$ makes the score of these points indistinguishable, and ($iii$) as the dimension of data increases, an increasing percentage of points will appear at the edge of the convex hull covering the data set leading to have low scores to every points.

To remedy those drawbacks, alternative depth functions have been independently introduced in Chen et al. (2015) and Ramsay et al. (2019). In this regard, the extension of Tukey's halfspace depth, recently introduced and referred to as the halfspace-mass (HM) depth (Chen et al., 2015) (see also Ramsay et al., 2019 and Staerman et al., 2021b), offers many advantages.

Authors proposed to replace the infimum by an expectation over all possible closed halfspaces containing $x$, following in the footsteps of Cuevas & Fraiman (2009). More precisely, given a random variable $X$ following a distribution $P$ and a probability measure $Q$ on $\mathcal{H}(x)$, it is defined as follows:

$$D_{\mathrm{HM}}(x, P) = \mathbb{E}_{H \sim Q} \left[ P(H) \right]. \tag{4}$$

In addition to basic properties a depth function should satisfy, the halfspace-mass depth possesses robustness properties: it has a unique (depth-induced) median with an optimal breakdown point equal to 0.5 (Chen et al., 2015) which means that the halfspace-mass depth provides a stable ordering of the 'normal' data

even when polluted data belong to the training set. In addition, it has been successfully applied to anomaly detection in Chen et al. (2015) making it a natural choice to adversarial attack detection. When a training set $\{x_i, \ldots, x_n\}$ is given, Eq. 4 boils down to:

$$D_{\text{HM}}(x, P_n) = \mathbb{E}_Q\left[\frac{1}{n}\sum_{i=1}^{n}\mathbb{I}\{x_i \in H\}\right],$$ (5)

where $P_n$ is the empirical measure defined by $\frac{1}{n}\sum_{i=1}^{n}\delta_{x_i}$. The expectation can be conveniently approximated with a Monte-Carlo scheme in contrast to several depth functions that are defined as solutions of optimization problems, possibly unfeasible in high dimension. The aim is then to approximate Eq. 5 by drawing a finite number of closed halfspaces containing $x$ (see Appendix A for the approximation algorithm for training and testing).

## 3.2 Our Depth-Based Detector

The methodology we propose here is based on the halfspace-mass depth that exhibits attractive mathematical and computational properties, as described in the previous section.

Our depth-based detector, HAMPER, relies on the information available in a subset $\Lambda$ of DNNs' layers, i.e., the mapped data $z_{\ell,i} = f_\theta^\ell(x_i; \mathcal{D})$, $\ell \in \Lambda \subset \{1, \ldots, L-1\}$. We denote by $\widetilde{\mathcal{S}}^\ell = \{z_{\ell,i}\}_{i=1}^{n}$ and $\widetilde{\mathcal{S}}_c^\ell = \left\{z_{\ell,i} \in \widetilde{\mathcal{S}}^\ell : y_i = c\right\}$, the $\ell$-th and the $\ell$-th class-conditionally representations of the training dataset, respectively. Our approach aims to construct a score function $s : \mathbb{R}^d \to [0, 1]$ providing a confidence level to a new observation $x$ indicating its degree of abnormality w.r.t. to the training dataset. HAMPER leverages appealing properties of the HM depth detailed in Sec. 3.1 and can be summarized into two distinct steps. The function $s$ is built by first constructing $|\Lambda| \times C$ intermediate scoring functions $s_{\ell,c} : \mathbb{R}^{d_\ell} \to [0, 1]$ designed for each considered layer and each class. The map $s_{\ell,c}$ assigns a value to any element of the embedded space of the $\ell$-th layer representing somehow its 'distance' to the class $c$ of the mapped training set. Thereafter, an aggregation is performed between scores using a small validation dataset composed of both 'normal' and 'adversarial' samples. These two parts of the proposed approach are detailed below.

**Intermediate score functions.** Given a new observed image $x \in \mathbb{R}^d$ mapped into $|\Lambda|$ representations $\{z_\ell\}_{\ell \in \Lambda}$ such that $z_\ell = f_\theta^\ell(x; \mathcal{D})$, we propose to use the HM depth as intermediate scoring functions $s_{\ell,c}$. Precisely, we compute $D_{\text{HM}}(z_\ell, \widetilde{\mathcal{S}}_c^\ell)$, for each considered layer $\ell$ and each class $c$, i.e., the HM depth between $z_\ell = f_\theta^\ell(x; \mathcal{D})$ and the class-conditionally probability distribution of the training dataset $\widetilde{\mathcal{S}}_c^\ell = \{f_\theta^\ell(x_i; \mathcal{D}) : x_i \in \mathcal{S}_c\}$. Following the approximation algorithm of the HM introduced in Chen et al. (2015), we use an efficient training/testing procedure in order to compute $D_{\text{HM}}$ (summarized in Algorithms 1 and 2 in Appendix A). These algorithms are repeated for each class $c$ and each considered layer $\ell$ leading to $|\Lambda| \times C$ scoring functions. Three parameters with low sensitivity are involved: $K$ which is the number of sampled halfspaces in order to approximate the expectation of Eq. 4; the size $n_s$ of the sub-sample drawn at each projection step; and the $\lambda$ hyperparameter which controls the extent of the choice of the hyperplane. In this paper, we follow the advice given in Chen et al. (2015) by choosing the following parameters $K = 10000$, $n_s = 32$ and $\lambda = 0.5$ offering a good compromise between performance and computational efficiency.

**Aggregation procedure.** Following the supervised setting scenario, as in Kherchouche et al. (2020); Feinman et al. (2017) or Ma et al. (2018), the score is obtained through an aggregation which is performed between halfspace-mass scores using a small validation dataset composed of 'normal' and 'adversarial' samples. Our scoring function is then formally defined as:

$$s(x) = \sum_{\ell \in \Lambda}\sum_{c=1}^{C}\alpha_{\ell,c}\, D_{\text{HM}}(z_\ell, \widetilde{\mathcal{S}}_c^\ell),$$ (6)

where the weights $\alpha_{\ell,c}$ are obtained through the training of a linear regressor in a supervised manner. It is worth noting that the anomaly score $s$ from Eq. 6 results from both class and layer dependent linear

combination. The class dependency of $s$ is motivated by (1) the per class behavior classifier which is highlighted in Sec. 4.2 and by (2) the monotonicity relative to deepest point property of the halfspace-mass depth and depth functions in general (see e.g. property ($\mathbf{D}_3$) in Appendix B or in Zuo & Serfling, 2000 and Staerman, 2022). The layer dependency of $s$ is motivated by the per layer behavior classifier which is displayed in Sec. 4.3.

Referring to the problem formulation notations (see Sec. 2.1), given a threshold $\gamma$, and supposing $a = 1$, the detector is provided by the Eq. 3. The overview of HAMPER can be summarized in the Algorithm 3 in Appendix A.

### 3.3 Comparison to state-of-the-art detection methods.

We benchmark our approach with three supervised detection methods: LID, KD-BU, and NSS. We chose these baselines because they are supervised and do not modify the model to protect. We could also consider Abusnaina et al. (2021) but unfortunately the code is not publicly available. We provide the comparison to other two other state-of-the-art methods in Appendix E, Mahalanobis (Lee et al., 2018) and JTLA (Raghuram et al., 2021).

**Local Intrinsic Dimensionality (LID).** LID is based on the intuition that adversarial examples lie outside of the clean data manifold. By computing the Local Intrinsic Dimensionality, it is possible to check whether the new point is close to the original data manifold. Following this idea, three version of each natural samples are used. The clean one, a noisy version of it, and an attacked one. The LID approximate for each of those points to the clean distribution will be computed at the output of each layers. Those variables will then be used to train a detector that will distinguish between adversarial samples, and normal ones (normal samples are either clean or noisy samples). Each of the strategy to craft adversarial samples can have very different LID characteristics, a detector per type of threats is therefore necessary. LID therefore lies in the *attack-aware* category.

**Kernel Density and Bayesian Uncertainty (KD-BU).** KD-BU also relies on the idea that adversarial samples lie off the original data manifold. To detect adversarial samples, they first perform a kernel density estimate at the last hidden layer level to detect samples that are far from the original manifold. Then, a bayesian uncertainty estimate is computed to detect when points lie in low-confidence regions of the input space. Both of those characteristics are later used to train a detector that distinguish between natural and adversarial examples. Once again, the kernel density estimates and the bayesian uncertainty values for different types of attacks can differ a lot, therefore this method have been created to be *attack-aware*.

**Natural Scene Statistics (NSS).** NSS relies on the extraction of the natural scene statistics at the image level. Natural scene statistics are statistics that will be very different for natural and attacked images. Indeed, for clean image, applying the natural scene statistics will output an image with meaning, however, for attacked samples, the resulting image will have no meaning. The Natural Scene Statistics extraction is then used to train a detector to distinguish between natural and attacked samples. To overcome the need to have a specific detector per attack, the authors of NSS decided to train their detector using the natural scene statistics of various attacks. It therefore lies in the *blind-to-attack* category.

**HAMPER.** Our proposed HAMPER detector is computing, thanks to the halfspace-mass depth, the distance of a given $x$ to a reference training distribution. In the sense that it compares between a novel point and a reference, our method is close to the LID method, however, as explained in the previous section, we do not compute our anomality score (i.e., the HM depth) at each layer's output, but we only use a subset of layers $\Lambda$. In addition, it is possible to use our proposed detector under both scenarios, i.e., the *attack-aware* and the *blind-to-attack* scenarios.

## 4 Analyzing Statistical Information of the Networks' Behavior under Threats

In this section, we provide insights into the attackers' and defenses' behavior from the classifier's perspective. This section aims to provide justification and insights on the choice of making the linear weights of Eq. 6 dependant on both the class and the layer. In particular, we analyze the behavior of the classifier on attacks

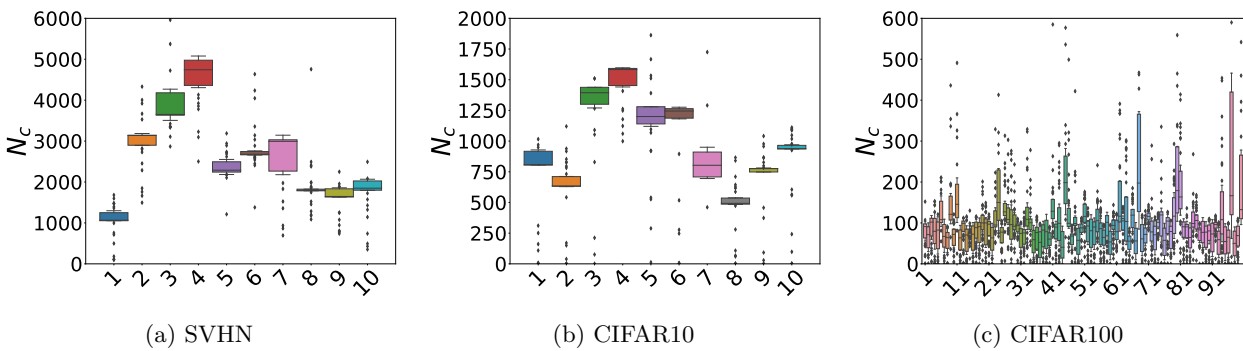

(a) SVHN         (b) CIFAR10         (c) CIFAR100

Figure 2: **Per class behavior analysis.** Average number of adversarial examples per class on each of the considered datasets.

in Sec. 4.2, while in Sec. 4.3 we explore which subset of layers of the classifier carries the relevant information to build an efficient supervised data-depth based detector.

## 4.1 Experimental setting

**Datasets and classifiers**. We run our experiments on three image datasets: SVHN (Netzer et al., 2011), CIFAR10 and CIFAR100 (Krizhevsky, 2009). We train a classifier that aims at rightfully classifying natural examples for each of those datasets. For SVHN and CIFAR10 we use a ResNet-18 trained for 100 epochs, using an SGD optimizer with a learning rate of 0.1, weight decay of $10^{-5}$, and a momentum of 0.9; for CIFAR100 we chose a ResNet-110 pre-trained[1] using an SGD optimizer with a learning rate of 0.1, weight decay of $10^{-5}$, and a momentum of 0.9. Once trained, all classifiers are frozen.

**Attacks & choice of the maximal allowed perturbation $\varepsilon$.** To have a wide range of attacks to test, we use all the methods mentioned in Sec. 2.3. For FGSM, BIM and PGD, we consider the $L_\infty$-norm, with multiple $\varepsilon$ in {0.0315, 0.0625, 0.125, 0.25, 0.3125, 0.5}. We also generate perturbed examples using PGD under the $L_1$-norm constraint with $\varepsilon$ varying in {5, 10, 15, 20, 25, 30, 40}, for CIFAR10 and SVHN, and in {40, 500, 1000, 1500, 2000, 2500, 5000} for CIFAR100. Moreover, we generate perturbed examples using PGD under the $L_2$-norm constraint with $\varepsilon$ varying in {0.125, 0.25, 0.3125, 0.5, 1, 1.5, 2}, for CIFAR10 and SVHN, and in {5, 10, 15, 20,

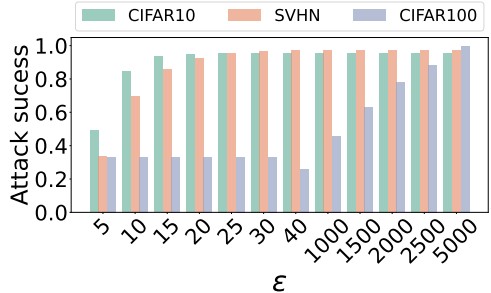

Figure 3: **Calibrating the maximal allowed perturbation $\varepsilon$ on CIFAR100**. Accuracy on adversarial examples created using $PGD_1$ for the SVHN, CIFAR10 and CIFAR100 classifiers. *On CIFAR100, to ensure high successes of the attacks, one must allow the attacker to have larger values of $\varepsilon$, compared to the CIFAR10 and SVHN ones.*

30, 40, 50} for CIFAR100. In order to attack with PGD ($L_1$ and $L_2$ norm) the classifier trained on CIFAR100, we chose different epsilon values than those used for CIFAR10 and SVHN since the attacks generated with those epsilons were not able to fool the network (see Fig. 3). CW attacks are generated under the $L_\infty$ and $L_2$ constraint, with $\varepsilon$ equals to 0.3125 and 0.01 respectively. Finally, we perturb samples using DF which is an $L_2$ attack without any constraint on $\varepsilon$. Concerning blackbox attack, SA is an $L_\infty$-norm attack $\varepsilon = 0.125$, HOP is an $L_2$ attack with $\varepsilon = 0.1$. Finally, ST is not concerned by a norm constraint nor a maximal perturbation, the attacker strength is limited in rotation (maximum of $30^o$) and in translation (maximum of 8 pixels).

---

[1] https://github.com/bearpaw/pytorch-classification

### 4.2 Analyzing the Networks' per Class Behavior under Threats

In this section, we investigate the per-class behavior of the image classifier to motivate and justify the choice of *the class dependency of the proposed aggregation procedure* (see Eq. 6).

**Simulation.** We examine the distribution of the adversarial examples w.r.t. the class predicted by the classifier. For this purpose, in Fig. 2, we plot the distribution of adversarial samples per predicted class ($N_c$) as a function of the class ($c$).

**Analysis.** In SVHN, CIFAR10 and CIFAR100 natural images are balanced. However, on these datasets, the per-class distribution of the adversary is not uniform over the classes: in both SVHN (Fig. 2a) and CIFAR10 (Fig. 2b), classes 3 and 4 are overly represented on average, on the contrary of class 1 and 8 for SVHN and CIFAR10 respectively. Similarly, in CIFAR100 (Fig. 2c), the classes 11, 24, 45, 68, 80, 81, and 97 are the overrepresented whilst the classes 37, 59, 65, 67, 76 and 98 are the most underrepresented on average. Note that the diamond points in the plots denote the outliers, i.e., adversarial examples behaving differently from the others.

**Takeaways.** The variability of the per-class behavior of the classifier under threats suggests that class is an important characteristic and should be leveraged to detect adversaries. This observation further motivates the per-class computation of the halfspace-mass depths and then the class dependency of the linear regressor of Eq. 6.

### 4.3 Analyzing the Networks' per Layer Behavior under Threats

In this section, we investigate the per-layer behavior of the image classifier to motivate and justify the choice of *the layer behavior of the proposed aggregation procedure* (see Eq. 6).

**Simulation.** We investigate each layer's roll on HAMPER 's decision process to better understand each layer importance. In particular, we focus on CIFAR10 with ResNet-18 and we train a linear least squares regressor with $L_2$ regularization (see e.g. Hastie et al., 2009) on the depth features extracted from all the layers, $\Lambda \in \{1, \ldots, L-1\}$, of the classifier. In Fig. 4, we report the weights associated with each layer $\ell \in \Lambda$ of the underlying classifier, averaged over the classes, when changing the values of the $L_2$ weight constraint (the values are reported in the legend).

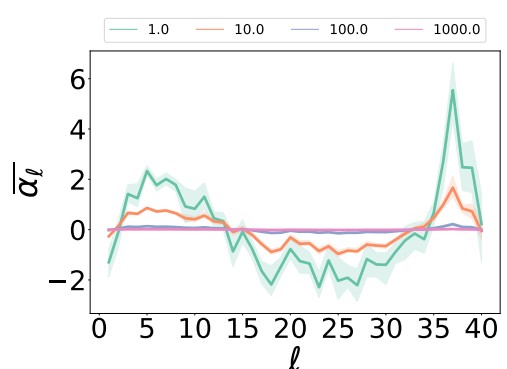

Figure 4: **Per layer behavior analysis.** Evolution of $\bar{\alpha}_\ell = \frac{1}{C} \sum_c \alpha_{\ell,c}$ the average and the standard deviation over the classes of the regressor weight as a function of the layer, for different value of the regularization parameter.

**Analysis.** From Fig. 4, we observe of the decision of the detector is based on several layers. This a-posteriori analysis justifies the layer dependency of the linear regressor weights. As the $L_2$ weight regularization increases, the last layers receive more weights, suggesting they are good candidates to build a detector. Although it is possible to rely on the whole set of layers, motivated by efficiency and Fig. 4, we select the 11 final layers of the classifier, i.e., we base our experiments on the subset $\Lambda = \{L - 12, \ldots, L - 1\}$.

**Takeaways.** Through Fig. 4, we have motivated both the per-layer computation of the halfspace-mass depths and then the per-layer dependency of the linear regressor of Eq. 6.

## 5 Experiments

In this section, we assess the effectiveness of our proposed depth-based detection method. The code is available at https://github.com/MarinePICOT/HAMPER. This section is organized as follows: we first describe the experimental setting in Sec. 5.1 and then we provide a detailed discussion of the results in Sec. 5.2.

Table 1: Attack-aware performances on the three considered datasets - SVHN, CIFAR10 and CIFAR100 - of HAMPER$_{\text{AA}}$ detector together with the results of the SOTA detection methods: LID, KD-BU, and averaged over the L$_p$-norm constraint. The best results among the detectors are shown in **bold**. The results are presented as AUROC↑ (FPR ↓$_{95\%}$ %) and in terms of mean ($\mu$) and standard deviation ($\sigma$).

| | | LID | | | KD-BU | | | HAMPER$_{\text{AA}}$ | | |
|---|---|---|---|---|---|---|---|---|---|---|
| | | SVHN | CIFAR10 | CIFAR100 | SVHN | CIFAR10 | CIFAR100 | SVHN | CIFAR10 | CIFAR100 |
| **Norm L$_1$** | $\mu$ | 64.9 (90.1) | 57.9 (87.7) | 77.6 (47.4) | 84.7 (58.7) | 71.0 (68.3) | 69.5 (71.2) | **100** (0.0) | **100** (0.0) | **100** (0.0) |
| | $\sigma$ | 9.2 (5.7) | 11.3 (8.6) | 23.4 (35.7) | 7.0 (16.5) | 24.1 (33.4) | 21.1 (31.1) | **0.0** (0.0) | **0.0** (0.0) | **0.0** (0.0) |
| **Norm L$_2$** | $\mu$ | 78.3 (73.7) | 66.7 (77.3) | 66.5 (65.5) | 84.3 (42.3) | 70.5 (62.2) | 56.2 (87.6) | **100** (0.0) | **100** (0.0) | **100** (0.0) |
| | $\sigma$ | 10.0 (16.9) | 15.0 (22.3) | 24.7 (32.8) | 16.2 (31.3) | 26.9 (40.9) | 17.0 (10.3) | **0.0** (0.0) | **0.0** (0.0) | **0.0** (0.0) |
| **Norm L$_\infty$** | $\mu$ | 97.5 (9.9) | 94.2 (21.2) | 66.9 (79.1) | 87.3 (21.9) | 93.1 (20.3) | 67.1 (80.3) | **100** (0.0) | **100** (0.0) | **100** (0.0) |
| | $\sigma$ | 3.7 (14.0) | 7.6 (24.0) | 17.3 (17.3) | 27.4 (29.9) | 16.8 (32.6) | 15.4 (11.3) | **0.0** (0.0) | **0.0** (0.0) | **0.0** (0.0) |
| **No Norm** | $\mu$ | 99.1 (4.4) | 91.7 (36.6) | 98.4 (4.2) | 92.8 (21.9) | 81.4 (76.2) | 76.1 (61.3) | **100** (0.0) | **100** (0.0) | **100** (0.0) |
| | $\sigma$ | **0.0** (0.0) | **0.0** (0.0) | **0.0** (0.0) | **0.0** (0.0) | **0.0** (0.0) | **0.0** (0.0) | **0.0** (0.0) | **0.0** (0.0) | **0.0** (0.0) |
| **Average** | $\mu$ | 86.5 (41.3) | 79.7 (48.6) | 77.4 (49.0) | 86.2 (34.1) | 82.8 (41.6) | 64.9 (80.1) | **100** (0.0) | **100** (0.0) | **100** (0.0) |
| | $\sigma$ | 14.8 (38.2) | 18.7 (36.6) | 21.4 (30.4) | 21.8 (31.5) | 23.8 (41.5) | 17.7 (17.7) | **0.0** (0.0) | **0.0** (0.0) | **0.0** (0.0) |
| **Global** | $\mu$ | 81.2 (46.3) | | | 78.0 (51.9) | | | **100** (0.0) | | |
| | $\sigma$ | 19.7 (37.0) | | | 23.2 (37.6) | | | **0.0** (0.0) | | |

## 5.1 Experimental setting

We refer to Sec. 5.1 for the datasets, the classifiers and the attacks we considered for our evaluation.

**Evaluation metrics.** For each threat scenario, the performance is measured in terms of two metrics:

AUROC↑ (higher is better): the *Area Under the Receiver Operating Characteristic curve* (ROC; Davis & Goadrich, 2006) represents the relation between *True Positive Rate* (TPR) - i.e. adversarial examples detected as adversarial - and *False Positive Rate* (FPR) - i.e. natural samples detected as adversarial. As can be checked from elementary computations the AUROC↑ corresponds to the probability that an natural example has higher score than an adversary sample.

FPR at 95% TPR↓ or FPR ↓$_{95\%}$ (lower is better): represents the percentage of natural examples detected as adversarial when 95% of the adversarial examples are detected. The FPR ↓$_{95\%}$ is of high interest in practical applications.

*Remark.* The ideal classifier would reach 100% of AUROC↑ and 0% of FPR ↓$_{95\%}$.

## 5.2 Detecting adversarial examples

We recall from Sec. 2.2 that we distinguish between two settings in the supervised context: the *attack-aware scenario* and the *blind-to-attack scenario*. Therefore, we conduct two sets of experiments. In the attack-aware scenario, for each attack we train a detectors on a validation set - composed of the first 1000 samples of the testing set - and tested on the remaining samples. We refer here to HAMPER-Attack-Aware (HAMPER$_{\text{AA}}$) and we compare it with LID and KD-BU. In the blind-to-attack scenario, we train a unique detector and we test it on all the possible attacks. We refer here to HAMPER-Blind-to-Attack (HAMPER$_{\text{BA}}$) and we compare it with NSS. Note that, while our competitors assign to each input sample the probability of being adversarial, i.e., adversarial samples are labeled as 1, in HAMPER the detector outputs the depth score. This means that a high score corresponds to a deep sample, i.e., a natural one, hence clean samples are labeled as 1.

### 5.2.1 Attack-aware scenario

Here, we study the performance of the different detectors in the attack-aware scenario and show that HAMPER$_{\text{AA}}$ outperforms existing detectors.

Table 2: Blind-to-Attack detector performances on the three considered datasets - SVHN, CIFAR10, and CIFAR100 - of the HAMPER$_{\text{BA}}$ detector together with the results of the state-of-the-art detection methods, i.e., NSS, averaged over the L$_p$-norm constraint, along with the average and global performances. The best results among the detectors are shown in **bold**. The results are presented as AUROC↑ (FPR ↓$_{95\%}$ %) and in terms of mean ($\mu$) and standard deviation ($\sigma$).

| | | NSS | | | HAMPER$_{\text{BA}}$ | | |
|---|---|---|---|---|---|---|---|
| | | SVHN | CIFAR10 | CIFAR100 | SVHN | CIFAR10 | CIFAR100 |
| **Norm L$_1$** | $\mu$ | 69.2 (78.6) | 66.7 (80.1) | 69.7 (78.2) | **94.9** (24.9) | **94.7** (22.8) | **99.9** (0.2) |
| | $\sigma$ | 15.7 (24.3) | 10.3 (12.0) | 16.3 (32.0) | **3.1** (11.7) | **4.7** (19.6) | **0.0** (0.1) |
| **Norm L$_2$** | $\mu$ | 71.5 (65.8) | 68.0 (72.8) | 62.5 (90.2) | **94.2** (22.8) | **92.3** (30.2) | **99.9** (0.3) |
| | $\sigma$ | 19.5 (36.7) | 15.7 (26.8) | 9.4 (6.3) | **3.4** (12.6) | **7.2** (23.9) | **0.1** (0.2) |
| **Norm L$_\infty$** | $\mu$ | 93.6 (28.9) | 92.3 (15.5) | 69.1 (82.0) | **98.4** (7.3) | **98.7** (6.3) | **99.9** (0.3) |
| | $\sigma$ | 10.4 (43.5) | 20.1 (29.2) | 12.4 (13.6) | **2.4** (9.8) | **2.3** (10.7) | **0.0** (0.2) |
| **No Norm** | $\mu$ | **99.8** (0.4) | **93.8** (20.2) | 92.9 (24.7) | 98.5 (6.4) | 80.3 (57.1) | **100** (0.0) |
| | $\sigma$ | **0.0** (0.0) | **0.0** (0.0) | **0.0** (0.0) | 0.0 (0.0) | 0.0 (0.0) | **0.0** (0.0) |
| **Average** | $\mu$ | 83.5 (46.3) | 81.2 (42.3) | 68.1 (81.9) | **96.6** (14.6) | **95.8** (17.0) | **99.9** (0.3) |
| | $\sigma$ | 18.4 (43.6) | 21.2 (39.2) | 13.4 (20.1) | **3.4** (13.6) | **5.9** (20.9) | **0.1** (0.2) |
| **Global** | $\mu$ | 77.6 (56.8) | | | **97.4** (10.6) | | |
| | $\sigma$ | 19.2 (40.0) | | | **4.3** (16.2) | | |

**Global analysis.** We present the attack-aware evaluation of HAMPER$_{\text{AA}}$ together with LID and KD-BU on SVHN, CIFAR10 and CIFAR100. In Tab. 1, we group the results from Tab. 5 (that we relegate to Appendix C due to space constriction) according to the attack-norm (e.g., L$_1$, L$_2$, L$_\infty$, No norm), and we express them in terms of the mean on the AUROC↑ and the mean on the FPR ↓$_{95\%}$. We also report the average performances per dataset (*Average*) and over the datasets (*Global*).

In general, HAMPER$_{\text{AA}}$ outperforms the SOTA detectors by maintaining performance close to 100% AUROC↑ and 0% FPR ↓$_{95\%}$ on all the four datasets regardless of the attack-norm considered.

**Performance analysis per $\varepsilon$.** Overall, the results in Tab. 5 show that the smaller the perturbation magnitude $\varepsilon$ to craft the attack is, the more complex the attack detection. For example, the worst result of LID is with PGD1 and $\varepsilon = 15$ for SVHN and PGD1 with $\varepsilon = 20$ for CIFAR10 (AUROCs smaller than 50%). KD-BU exhibits the same attitude but for FGSM on SVHN, where it reaches its minimum (5.9% AUROC↑ and 99.5% FPR ↓$_{95\%}$) with $\varepsilon = 0.5$. Note that, the high value of the standard deviation in Tab. 1 could implies the detector is more susceptible to the $\varepsilon$ changes. This is particularly true in the case of the L$_1$-norm group since all the attacks considered are created with the same algorithm (PGD). On this regard, KD-BU turns out to be the detector most susceptible (e.g., on CIFAR10 its standard deviation on the FPR ↓$_{95\%}$ is 33.4 whilst the one of LID is 8.6 and the one of HAMPER$_{\text{AA}}$ is 0.0).

**Performance analysis per type of threat.** On average between the norm based attacks, LID and KD-BU more easily detects the L$_\infty$-norm attacks. Interestingly, with L$_\infty$ both LID and KD-BU have the best performance on CIFAR10 whilst on SVHN and CIFAR100 the detectors have the best performance in the no norm case (cf. Tab. 1). Consistently over the datasets, KD-BU poorly behaves on CW$_2$ as the AUROCs do not reach 50% (cf. Tab. 5).

**Summary.** Tab. 1 suggests LID and KD-BU have similar behaviors on SVHN, whilst KD-BU improves on CIFAR10. A closer look at Tab. 5 also suggests KD-BU has higher variance in the results w.r.t. LID on all the three datasets. Thus KD-BU performances are most affected by the perturbation magnitude to craft the adversarial examples. HAMPER$_{\text{AA}}$, on the other side, is hereby confirmed as the best detector since it does not change its performances no matter the perturbation or the norm considered in the attacks.

### 5.2.2 Blind-to-attack scenario

Here, we study the performance of the different detectors in the blind-to-attack scenario and show that HAMPER$_{\text{BA}}$ outperforms existing detectors.

**Global analysis.** We present the blind-to-attack evaluation of HAMPER$_{\text{BA}}$ together with NSS on SVHN, CIFAR10, CIFAR100. As in Sec. 5.2.1, in Tab. 2, we group the results from Tab. 6 according to the attack-norm and we express them in terms of mean / standard deviation on the AUROC↑ and mean / standard deviation on the FPR $\downarrow_{95\%}$. Moreover we report the average performances per dataset (*Average*) and over all the dataset (*Global*). On average, HAMPER$_{\text{BA}}$ outperforms NSS by 13.1(31.7) on SVHN, 14.6(25.3) on CIFAR10 and 31.8(81.6) on CIFAR100 in terms of AUROC↑(FPR $\downarrow_{95\%}$). Under L$_1$, L$_2$ and L$_\infty$-norm constraints, HAMPER$_{\text{BA}}$ outperforms NSS on all considered datasets. The increase goes up to 30.2(78.0) in L$_1$-norm, 37.4(89.9) in L$_2$-norm and 30.8(81.7) in L$_\infty$-norm in terms of in AUROC↑(FPR $\downarrow_{95\%}$).

**Performance analysis per $\varepsilon$.** Overall, the results in Tab. 6 show that, in contrast to HAMPER$_{\text{BA}}$, NSS' performances are increasing with the value of maximal allowed perturbation $\epsilon$. As a matter of fact, the performances of HAMPER$_{\text{BA}}$ on L$_1$ and L$_2$-norm constraints first decrease as $\varepsilon$ increases ($\varepsilon \in [5, 15]$ for L$_1$-norm constraint, and $\varepsilon \in [0.125, 0.3125]$ for L$_2$-norm constraint), until it starts increasing. On L$_\infty$-norm constraint, our method follows the expected behavior, i.e., the performances increases with $\varepsilon$.

**Performance analysis per type of threat.** Tab. 2 suggests that both HAMPER$_{\text{BA}}$ and NSS are globally better at detecting attacks with L$_\infty$-norm constraints, particularly those created with PGD and BIM as an attack strategy. However, on SVHN NSS finds more difficult to spot the FGSM-based attacks. SA and DeepFool threats are the thoughest to detect for NSS. On the contrary, while HAMPER$_{\text{BA}}$ consistently detect SA-based attack, it shows a slight drop in performance for DeepFool-based attacks on CIFAR10. Finally, NSS presents poor performances at detecting CW$_2$ attacks, while it is not the case for our proposed method.

**Summary.** While NSS's performances vary with the dataset, the $\varepsilon$ and the norm used to construct the attacks, HAMPER$_{\text{BA}}$ consistently detect them. In particular, we note that HAMPER$_{\text{BA}}$ is well suited to larger datasets (e.g., CIFAR100). Conversely, NSS' performance highly decreases when passing from the datasets with 10 classes to the one with 100.

### 5.3 Attacking HAMPER using Adaptive Adversary

The importance of attacking defenses with adaptive attacks has increased recently (Abusnaina et al., 2021; Raghuram et al., 2021). As mentioned in Sec. 2.3, the Backward-Pass Differentiable Attack (*BPDA*; Athalye et al., 2018a) are based on the possibility to find a suitable surrogate to the non-differentiable parts of any defense. However, deriving a suitable differentiable surrogate of the halfspace-mass depth remains an open research question which has never been tackled. As a matter of fact, the only attempt to approximate a non-differentiable depth was performed on the Tukey depth in She et al. (2021), with very poor results (Dyckerhoff et al., 2021). It is worth to point out that, if approximating the Tukey depth is already hard, finding a differentiable surrogate for IRW would be even harder because it involves several non-differentiable components (the indicator function, the function $x \mapsto \min\{x, 1 - x\}$ and the need to compute a ranking). Finding a differentiable suitable surrogate to attack HAMPER would, therefore, require a substantial effort and should be rigorously handled. As a consequence, we have to rely on adaptive blackbox attackers, as suggested in Tramer et al. (2020), to attack HAMPER.

**Experimental Setting.** We designed a blackbox adaptive attacker by using SA, which is both effective and computationally efficient (Engstrom et al., 2019). To extend SA into an adaptive attack, we modified the success criterion to not only fool the classifier but also the detector, and the loss, which becomes a trade-off between minimizing the difference between the logits of the two most probable classes and maximizing the detector's prediction. In Tab. 3, we present the results of the adaptive SA attack on CIFAR10 for both NSS and HAMPER$_{\text{BA}}$, where $\alpha$ controles the trade-off between the two objectives of the attack.

**Analysis.** On Tab. 3, we varied the value of the parameter $\alpha$ controlling the trade-off between the classifier and the detector performances. As $\alpha$ increases, the performance of the attack on the classifier decreases while the detector has more and more trouble detecting them, as one can expect. On the considered adaptive

Table 3: Detector performances and Attack's success under adaptive blackbox attacks for NSS and HAMPER$_{BA}$ on CIFAR10. We present the results as: AUROC↑% (FPR ↓$_{95\%}$ %) for the detector performances.

| | **Adaptive Attacks** | | | |
| | CIFAR10 | | | |
| | NSS | | HAMPER$_{BA}$ | |
| | Detector Performances | Attack Success (%) | Detector Performances | Attack Success (%) |
|---|---|---|---|---|
| $\alpha = 10^{-3}$ | 9.4 (100) | 95.82 | **98.5** (8.2) | 98.41 |
| $\alpha = 10^{-2}$ | 7.2 (100) | 96.12 | **96.9** (21.6) | 99.37 |
| $\alpha = 10^{-1}$ | 5.5 (100) | 93.60 | **84.0** (60.8) | 98.41 |
| $\alpha = 1$ | 7.8 (100) | 95.88 | **91.2** (50.5) | 99.12 |
| $\alpha = 10^1$ | 3.2 (100) | 93.28 | **72.5** (70.7) | 96.60 |
| $\alpha = 10^2$ | 3.5 (100) | 93.83 | **75.1** (71.0) | 72.80 |
| $\alpha = 10^3$ | 3.7 (100) | 93.44 | **64.9** (81.6) | 28.24 |

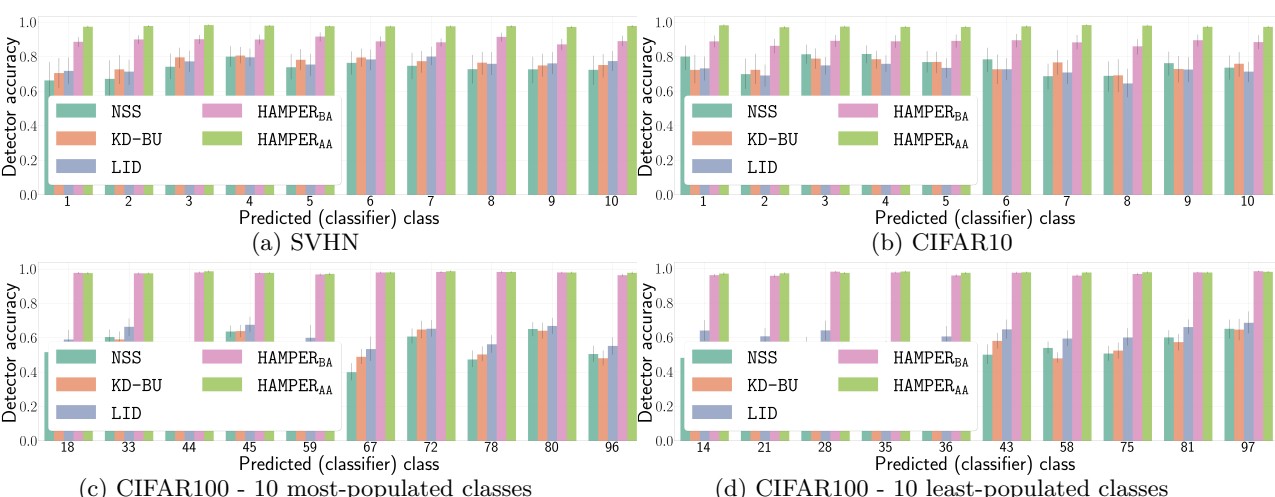

Figure 5: **Accuracies of the detectors per predicted classifier class**. For visualization reasons, we restrict the plot to the 10-most/least populated classes for CIFAR100.

attacks, it is clear that it is easy for the attacker to find powerful and undetected samples to attack NSS. However, it is more difficult to fool our proposed method. To decrease the AUROC↑ to a value close 50 (which is equivalent to a random detector), the attacker is only able to fool the classifier 28% of the time.

**Takeaways.** HAMPER$_{BA}$ is more robust to adaptive attacks than NSS.

## 5.4 Further Analysis of the Detector Behaviors

In this section, we first investigate the per-class behavior of the detectors to further asses the effectiveness of the proposed method (cf. Sec. 5.4.1). Then, we study the AUROC↑/FPR ↓$_{95\%}$ trade-off (cf. Sec. 5.4.2) and the time and resource constraints (cf. Sec. 5.4.3) for each of the considered methods.

### 5.4.1 Analyzing the detectors' per-class behavior

Sec. 4.2 identifies a class-dependant behavior of the attack mechanisms that could translate in a class-dependant behavior of the different detection methods. In this experiment, we study the performance distribution per class.

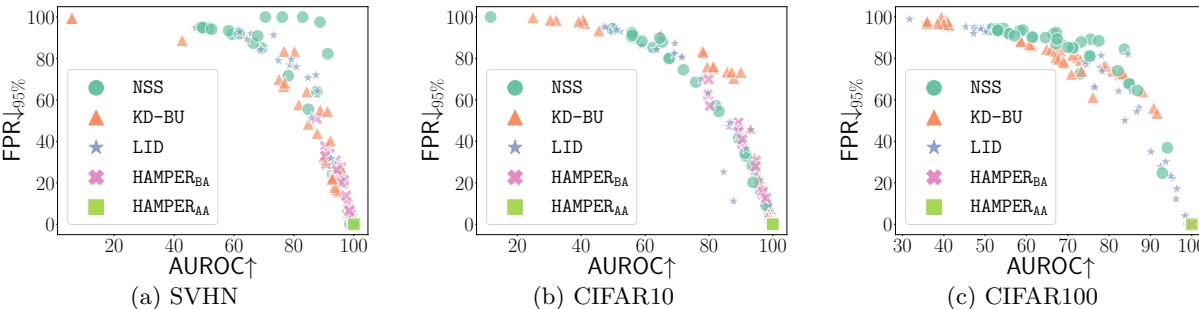

Figure 6: FPR$\downarrow_{95\%}$ as a function of the AUROC$\uparrow$ for all five considered methods, (a) on SVHN, (b) on CIFAR10, and (c) on CIFAR100. This figure aggregates all the results from Tab. 5 and Tab. 6.

**Simulation.** In Fig. 5, we examine the accuracy of the detectors on the testing samples (natural and adversarial) w.r.t. the class predicted by the classifier. Note that, for visualization purposes, we select the 10-most/least populated classes in the case of CIFAR100. For this simulation, we select the threshold $\gamma$ for which TPR is at 95%.

**Analysis.** HAMPER performances are not affected either by the class nor by the dataset and they consistently outperform the competitors. However, the competitor's performance are class-dependant: they tend to better distinguish between natural and attacked samples for classes with the highest number of adversarial examples. This is demonstrated by Fig. 5a where NSS, KD-BU and LID obtain the highest accuracy in class 4 on SVHN, which is also the most populated class (cf. Fig. 2a). A similar behavior is observable in CIFAR10 (cf. classes 3 and 4 in Fig. 5b and Fig. 2b). Moreover, Fig. 5 suggests that, in terms of accuracy per class, the SOTA methods show similar performances on SVHN; on the contrary, on CIFAR10 the detector performing the best is NSS while on CIFAR100 it is LID.

**Takeaways.** *Differently from the competitors, the HAMPER detectors are not affected by the per-class distribution of the samples.* In particular, and regardless of the dataset, the proposed detectors show a uniform behavior overall the classes. Further confirmation is given from the plots in Fig. 5c and Fig. 5d where the accuracies of the detectors remain constant even when focusing on only the most and the least-populated classes respectively.

### 5.4.2 Studying the AUROC↑-FPR $\downarrow_{95\%}$ relationship

As commonly done in anomaly detection, we measure the detection performances in terms of AUROC$\uparrow$ and FPR $\downarrow_{95\%}$. However, to a large AUROC$\uparrow$ does not necessarily correspond a low FPR $\downarrow_{95\%}$. In this experiment, we study the trade-off between AUROC$\uparrow$ and FPR $\downarrow_{95\%}$ performances for each considered detection method.

**Simulation.** In Fig. 6, we analyze the trade-off between AUROC$\uparrow$ and FPR $\downarrow_{95\%}$. We translate on the Cartesian planes the results presented in Tab. 5 and in Tab. 6. The perfect detector will have the points in (100, 0).

**Analysis.** Interestingly, the HAMPER detectors behave closely to the perfect detector for all the considered attacks. This is particularly true for HAMPER$_{AA}$. For NSS, KD-BU and LID, a large AUROC$\uparrow$ does not necessarily correspond a low FPR $\downarrow_{95\%}$. In SVHN (Fig. 6a), five of the NSS points are exhibiting high AUROC$\uparrow$ (between 70% and 92%) while presenting extremely high FPR $\downarrow_{95\%}$ (between 81% and 99.6%). A similar behavior is presented in CIFAR10 with LID.

**Takeaways.** Contrary to other detection method, which can exhibit high FPR $\downarrow_{95\%}$ for high AUROC$\uparrow$, our proposed detectors behave similarly to the perfect detector for both attack-aware and blind-to-attack scenarios.

### 5.4.3 Time and resources constraints

For some applications, time and resource necessity can be critical. We, therefore, decided to measure the constraints of each considered method.

**Simulations.** In Tab. 4, We report the time and resource constraints needed for each method.

**Analysis.** All methods have quite comparable training time. However, `LID`, due to the extraction of the `LID` parameters to all layers, takes a lot more time to test than the others.

**Takeaways.** `HAMPER`'s deployment requires comparable time and resources to the other considered detectors.

Table 4: Time and computational constraints to train and test each detection method. Reported times include all required steps for each methods.

| Method | GPUs | Training Time | Testing Time |
|---|---|---|---|
| NSS | V100-16G | 00m30s | 00m55s |
| KD–BU | V100-16G | 00m30s | 02m00s |
| LID | V100-16G | 04m00s | 35m00s |
| HAMPER$_{\text{AA}}$ | V100-16G | 02m00s | 02m00s |
| HAMPER$_{\text{BA}}$ | V100-16G | 02m00s | 02m00s |

## 6 Concluding Remarks and Future Work

In this paper, we introduced `HAMPER`, a simple and effective method to detect adversarial attacks. One of the keys of `HAMPER` is to rely on the halfspace-mass depth, a statistical tool that remains overlooked by the machine learning community. Through our extensive experiments based on two scenarios, attack-aware, and blind-to-attack, we demonstrate that `HAMPER` achieves state-of-the-art performances. On average, it outperforms the existing best detector by 13.1% (29.0%) in terms of AUROC↑ (resp. FPR $\downarrow_{95\%}$) on SVHN, 14.6% (25.7%) on CIFAR10, and 22.6% (49.0%) on CIFAR100. Interestingly, `HAMPER` exhibits class-independence, and less dependence on the norm-attack and the threat scenarios than other adversarial attack detection methods. As `HAMPER` relies on data depth that is not differentiable and finding surrogates is currently an open problem, our analysis relies on black-box adaptive attacker for which our method is more robust than existing methods.

Like all the supervised adversarial detection methods in the literature, `HAMPER` requires knowledge of the kind of attack evaluated at testing time, meaning that they are generally validated by assuming a single implicitly known attack strategy, which does not necessarily account for real-life threats. In future work, we will investigate how `HAMPER` and its main competitors perform when facing samples crafted through multiple unknown attack strategies at test time. We will mainly make an effort to understand whether the notion of depth is well suited to better generalize the detection of adversarial examples to attacks that are not involved in the supervised framework.

### Broader Impact Statement

Many concerns have been raised about the potential failures of Deep Learning: large neural networks are not trustworthy enough, limiting their adoption in high-risk applications. This paper's main contribution aims at improving the reliability of Deep Learning by designing a tool to prevent a malicious agent from disrupting the functioning of the system. Thus we believe our work will have a positive impact on society.

### Acknowlegments

This work has been supported by the project PSPC AIDA: 2019-PSPC-09 funded by BPI-France.

This work was performed using HPC resources from GENCI-IDRIS (Grant 2022-AD011013737 and 2022-AD01101838).

The work of Federica Granese was supported by the ERC project Hypatia under the European Unions Horizon 2020 research and innovation program. Grant agreement N. 835294.

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

# Supplementary Material

## A  Approximation algorithms

In this part, we present algorithms, originally proposed in Chen et al. (2015) and adapted to our problem, that are used in steps 3 and 4 in HAMPER (see Algorithm 1 for the training and Algorithm 2 for the testing). 3 shows the different steps to compute the scores for our HAMPER method

---

**Algorithm 1** Training algorithm for the approximation of $D_{\mathrm{HM}}$.

---

**Input**: sample $\widetilde{\mathcal{S}}_c^\ell = \{z_{\ell,i} \in \widetilde{\mathcal{S}}^\ell : y_i = c\}$.

**Initialization**: Number of halfspaces $K$; sub-sample size $n_s$; hyperparameter $\lambda$.

 1: **for** $k = 1, \dots, K$ **do**

 2:    Draw $\widetilde{\mathcal{S}}_{c,n_s}^\ell$, a sub-sample of $\widetilde{\mathcal{S}}_c^\ell$ with size $n_s$ without replacement.

 3:    Draw randomly and uniformly a direction $u_k$ in $\mathbb{S}^{d-1}$.

 4:    Compute $\langle u_k, z_{\ell,i} \rangle$ for every $z_{\ell,i} \in \widetilde{\mathcal{S}}_{c,n_s}^\ell$ such that $p_{k,i} \triangleq \langle u_k, z_{\ell,i} \rangle$.

 5:    Set $\mathrm{mid}_k = \left( \max_i p_{k,i} + \min_i p_{k,i} \right)/2$ and $\mathrm{range}_k = \max_i p_{k,i} - \min_i p_{k,i}$.

 6:    Randomly and uniformly select $\kappa_k$ in $\left[ \mathrm{mid}_k - \frac{\lambda}{2}\mathrm{range}_k, \ \mathrm{mid}_k + \frac{\lambda}{2}\mathrm{range}_k \right]$.

 7:    Set $m_k^{\mathrm{left}} = \dfrac{|\{z_{\ell,i} \in \widetilde{\mathcal{S}}_{c,n_s}^\ell : \ p_{k,i} < \kappa_k\}|}{n_s}$ and $m_k^{\mathrm{right}} = \dfrac{|\{z_{\ell,i} \in \widetilde{\mathcal{S}}_{c,n_s}^\ell : \ p_{k,i} \geq \kappa_k\}|}{n_s}$.

 8: **end for**

**Output**: $\{u_k, \kappa_k, m_k^{\mathrm{left}}, m_k^{\mathrm{right}}\}_{k=1}^K$.

---

**Algorithm 2** Testing algorithm for the approximation of $D_{\mathrm{HM}}$.

---

**Input**: test observation $z_\ell$; $\{u_k, \kappa_k, m_k^{\mathrm{left}}, m_k^{\mathrm{right}}\}_{k=1}^K$.

**Initialization**: HM=0.

 1: **for** $k = 1, \dots, K$ **do**

 2:    Project $z_\ell$ onto $u_k$ and such that $p_k^\ell = \langle z_\ell, u_k \rangle$.

 3:    $\mathrm{HM} = \mathrm{HM} + m_k^{\mathrm{left}} \, \mathbb{I}(p_k^\ell < \kappa_k) + m_k^{\mathrm{right}} \, \mathbb{I}(p_k^\ell \geq \kappa_k)$.

 4: **end for**

**Output**: $D_{\mathrm{HM}}(z_\ell, \widetilde{\mathcal{S}}_c^\ell) = \mathrm{HM}/K$.

---

---

**Algorithm 3** HAMPER

---

**Input**: test representations $\{z_\ell\}_{\ell=1}^{L-1}$; sample representations $\widetilde{\mathcal{S}}_c^\ell = \{z_{\ell,i} \in \widetilde{\mathcal{S}}^\ell : y_i = c\}$.

**Initialization**: Number of closed halfspaces $K$; sub-sample size $n_s$; hyperparameter $\lambda$.

1: **for** $\ell = 1, \dots, L-1$ **do**

2:      **for** $c = 1, \dots, C$ **do**

3:          Draw $K$ closed halfspaces containing $z_\ell$ using Algorithm 1.

4:          Compute $D_{\mathrm{HM}}(z_\ell, \widetilde{\mathcal{S}}_c^\ell)$ using Algorithm 2.

5:      **end for**

6: **end for**

7: Perform a linear regression to find weights $\alpha_{\ell,c}$ such that $s(x) = \sum_{\ell=1}^{L-1} \sum_{c=1}^{C} \alpha_{\ell,c}\, D_{\mathrm{HM}}(z_\ell, \widetilde{\mathcal{S}}_c^\ell)$.

**Output**: the score function $s$.

---

# B    Formal description of essential properties of Data Depths

Formally, a data depth function is defined as follows:

$$D : \quad \begin{matrix} \mathbb{R}^d \times \mathcal{P}(\mathbb{R}^d) & \longrightarrow & [0,1], \\ (x, P) & \longmapsto & D(x, P), \end{matrix} \qquad (7)$$

where $\mathcal{P}(\mathbb{R}^d)$ denotes the space of all probability distributions on $\mathbb{R}^d$. The higher $D(x, P)$, the deeper $x$ is in $P$. The depth-induced median of $P$ is then defined by the set attaining $\sup_{x \in \mathbb{R}^d} D(x, P)$ in the case where it exists. Since data depth naturally and in a nonparametric way defines a pre-order on $\mathbb{R}^d$ w.r.t. a probability distribution, it can be seen as a centrality-based alternative to the cumulative distribution function for multivariate data. Clearly, equation 7 opens the door to a variety of existing definitions Zuo (2003); Liu (1990); Cuevas et al. (2007); Koshevoy & Mosler (1997); Staerman et al. (2021b); Ramsay et al. (2019); Chen et al. (2015). While these differ in theoretical and practically related properties such as robustness or computational complexity, several postulates have been developed throughout the recent decades the "good" depth function should satisfy. Such properties have been thoroughly investigated in Liu (1990); Zuo & Serfling (2000) and Dyckerhoff (2004) with slightly different sets of axioms (or postulates) to be satisfied by a depth function. They are recalled below.

($\mathbf{D_1}$) (AFFINE INVARIANCE) Denoting by $P_X$ the distribution of any r.v. $X$ taking its values in $\mathbb{R}^d$, we have:
$$\forall x \in \mathbb{R}^d, \ \ D(Ax + b, P_{AX+b}) = D(x, P_X),$$
for any $d \times d$ nonsingular matrix $A$ with real entries and any vector $b$ in $\mathbb{R}^d$.

($\mathbf{D_2}$) (MAXIMALITY AT CENTER) For any $P \in \mathcal{P}(\mathbb{R}^d)$ that has a symmetry center $x^*$ (in a sense to be specified), the depth function $D(., P)$ takes its maximum value at it:
$$D(x^*, P) = \sup_{x \in \mathbb{R}^d} D(x, P).$$

($\mathbf{D_3}$) (MONOTONICITY RELATIVE TO DEEPEST POINT) For any $P \in \mathcal{P}(\mathbb{R}^d)$ with deepest point $x^*$, the depth at any point $x$ in $\mathbb{R}^d$ decreases as one moves away from $x^*$ along any ray passing through it:
$$\forall \xi \in [0,1], \ \ D(x^*, P) \geq D(x^* + \xi(x - x_P), P).$$

($\mathbf{D_4}$) (VANISHING AT INFINITY) For any $P \in \mathcal{P}(\mathbb{R}^d)$, the depth function $D$ vanishes at infinity:
$$D(x, P) \to 0, \ \text{as} \ ||x|| \to \infty.$$

These properties introduced in Zuo & Serfling (2000) lead to the following definition of a data depth function.

**Definition B.1.** A function $D : \mathbb{R}^d \times \mathcal{P}(\mathbb{R}^d) \longrightarrow [0, 1]$ is a statistical depth function if it satisfies $(\mathbf{D}_1 - \mathbf{D}_4)$.

**Discussion on properties.** The affine invariance property includes common transformations such as orthogonal, translation or scaling, and is useful in applications providing independence w.r.t. measurement units and coordinate system. For distributions having a uniquely defined center (e.g. symmetry center $x^*$), data depths should be maximized at this center, as stated by $(\mathbf{D}_2)$. The property $(\mathbf{D}_3)$ is a consequence of the center-outward ordering construction of data depth. When a point $x \in \mathbb{R}^d$ moves away from the set of elements that reach the maximum value of the depth function (potentially reduced to a single element, e.g. for symmetric distributions defined above), $D(x, P)$ should decrease monotonically.

**Intuition on the halfspace-mass depth.** The halfspace-mass depth is based on a simple idea. It aims to compute an anomaly score by exploring the rank of an element (w.r.t. the training data) on every possible one-dimensional projection. An intermediate score is then assigned to the element according to this rank. Further, the halfspace-mass depth value is the expectation of these intermediate (one-dimensional) anomaly scores. It is worth noting that a Monte-Carlo approximation replaces this expectation in practice by computing the average on a finite number of projections. The intuition is that an anomaly should be, on average, farther from the median than normal samples over all possible directions of $\mathbb{R}^d$. This depth function has the advantage of being simple, interpretable and computationally fast.

# C  Detailed Results on Perfect Knowledge about the attacker

Table 5: Performances on the fourconsidered datasets - SVHN, CIFAR10, CIFAR100 and Tiny ImageNet- of the HAMPER$_{AA}$ detector together with the results of the state-of-the-art detection methods: LID, and KD-BU, on multiple threat scenarios with multiple maximal perturbations $\varepsilon$. The best results among the detectors are shown in **bold**. The results are presented as:  AUROC↑ (FPR ↓$_{95\%}$ %). * stipulates the non-gradient based attacks.

| Norm L$_1$ | LID | | | | KD-BU | | | | HAMPER$_{AA}$ | | | |
|---|---|---|---|---|---|---|---|---|---|---|---|---|
| | SVHN | CIFAR10 | CIFAR100 | Tiny | SVHN | CIFAR10 | CIFAR100 | Tiny | SVHN | CIFAR10 | CIFAR100 | Tiny |
| PGD$_1$ | | | | | | | | | | | | |
| $\varepsilon = 5$ ($\varepsilon^\star = 40$) | 78.6 (76.9) | 63.5 (84.3) | 49.6 (94.1) | 37.2 (98.4) | 81.6 (57.8) | 32.2 (98.4) | 40.5 (96.3) | 61.3 (84.1) | **100** (0.0) | **100** (0.0) | **100** (0.0) | **100** (0.0) |
| $\varepsilon = 10$ ($\varepsilon^\star = 500$) | 66.6 (89.1) | 52.2 (92.9) | 36.0 (97.1) | 79.2 (68.8) | 75.0 (70.0) | 41.1 (96.9) | 40.2 (98.6) | 67.8 (79.2) | **100** (0.0) | **100** (0.0) | **100** (0.0) | **100** (0.0) |
| $\varepsilon = 15$ ($\varepsilon^\star = 1000$) | 47.2 (94.8) | 48.9 (93.7) | 76.2 (66.2) | 91.3 (39.7) | 76.7 (83.6) | 64.9 (91.4) | 64.9 (84.6) | 77.9 (71.8) | **100** (0.0) | **100** (0.0) | **100** (0.0) | **100** (0.0) |
| $\varepsilon = 20$ ($\varepsilon^\star = 1500$) | 61.6 (92.7) | 46.8 (95.0) | 90.1 (35.0) | 94.8 (27.0) | 84.3 (64.1) | 77.9 (83.7) | 72.3 (80.7) | 85.4 (59.4) | **100** (0.0) | **100** (0.0) | **100** (0.0) | **100** (0.0) |
| $\varepsilon = 25$ ($\varepsilon^\star = 2000$) | 65.7 (92.0) | 47.0 (95.0) | 95.4 (22.2) | 96.9 (17.2) | 88.8 (55.2) | 87.8 (70.5) | 80.9 (74.0) | 90.9 (47.7) | **100** (0.0) | **100** (0.0) | **100** (0.0) | **100** (0.0) |
| $\varepsilon = 30$ ($\varepsilon^\star = 2500$) | 61.6 (94.1) | 68.9 (82.5) | 96.4 (17.1) | 98.1 (9.4) | 91.2 (54.6) | 94.2 (32.0) | 88.1 (64.0) | 94.4 (34.1) | **100** (0.0) | **100** (0.0) | **100** (0.0) | **100** (0.0) |
| $\varepsilon = 40$ ($\varepsilon^\star = 5000$) | 73.2 (91.3) | 77.7 (70.2) | 99.8 (0.1) | 99.1 (4.1) | 95.3 (25.5) | 98.7 (5.0) | 99.7 (0.2) | 97.9 (12.4) | **100** (0.0) | **100** (0.0) | **100** (0.0) | **100** (0.0) |
| **Norm L$_2$** | SVHN | CIFAR10 | CIFAR100 | Tiny | SVHN | CIFAR10 | CIFAR100 | Tiny | SVHN | CIFAR10 | CIFAR100 | Tiny |
| PGD$_2$ | | | | | | | | | | | | |
| $\varepsilon = 0.125$ ($\varepsilon^\star = 5$) | 80.8 (76.0) | 63.6 (84.3) | 45.3 (95.2) | 99.9 (0.1) | 84.8 (48.5) | 30.3 (98.6) | 40.9 (96.0) | 99.9 (0.0) | **100** (0.0) | **100** (0.0) | **100** (0.0) | **100** (0.0) |
| $\varepsilon = 0.25$ ($\varepsilon^\star = 10$) | 74.8 (79.1) | 59.0 (86.3) | 39.7 (96.5) | **100** (0.0) | 76.6 (66.6) | 40.6 (98.6) | 41.3 (96.2) | **100** (0.0) | **100** (0.0) | **100** (0.0) | **100** (0.0) | **100** (0.0) |
| $\varepsilon = 0.3125$ ($\varepsilon^\star = 15$) | 68.9 (84.3) | 51.2 (93.2) | 35.5 (97.0) | **100** (0.0) | 76.9 (68.3) | 39.2 (97.8) | 40.9 (97.8) | **100** (0.0) | **100** (0.0) | **100** (0.0) | **100** (0.0) | **100** (0.0) |
| $\varepsilon = 0.5$ ($\varepsilon^\star = 20$) | 64.3 (89.3) | 47.7 (95.1) | 76.6 (66.1) | **100** (0.0) | 80.2 (83.5) | 78.2 (83.1) | 39.4 (99.9) | **100** (0.0) | **100** (0.0) | **100** (0.0) | **100** (0.0) | **100** (0.0) |
| $\varepsilon = 1$ ($\varepsilon^\star = 30$) | 79.1 (79.6) | 69.3 (87.2) | 83.8 (50.0) | **100** (0.0) | 95.4 (28.9) | 98.8 (3.6) | 65.9 (84.2) | **100** (0.0) | **100** (0.0) | **100** (0.0) | **100** (0.0) | **100** (0.0) |
| $\varepsilon = 1.5$ ($\varepsilon^\star = 40$) | 84.7 (70.7) | 89.2 (45.9) | 92.5 (27.8) | **100** (0.0) | 98.2 (5.7) | 99.9 (0.0) | 73.1 (79.2) | **100** (0.0) | **100** (0.0) | **100** (0.0) | **100** (0.0) | **100** (0.0) |
| $\varepsilon = 2$ ($\varepsilon^\star = 50$) | 87.3 (72.0) | 95.0 (23.9) | 94.9 (23.3) | **100** (0.0) | 99.2 (0.0) | 99.9 (0.0) | 82.6 (72.2) | **100** (0.0) | **100** (0.0) | **100** (0.0) | **100** (0.0) | **100** (0.0) |
| DeepFool* | | | | | | | | | | | | |
| No $\varepsilon$ | 93.2 (29.4) | 71.4 (81.1) | 96.2 (12.2) | - | 95.0 (15.9) | 85.7 (73.5) | 70.9 (72.5) | - | **100** (0.0) | **100** (0.0) | **100** (0.0) | - |
| CW$_2$* | | | | | | | | | | | | |
| $\varepsilon = 0.01$ | 61.9 (92.8) | 51.0 (94.0) | 31.7 (98.9) | - | 42.7 (88.8) | 45.5 (93.4) | 35.9 (96.8) | - | **100** (0.0) | **100** (0.0) | **100** (0.0) | - |
| HOP | | | | | | | | | | | | |
| $\varepsilon = 0.1$ | 87.8 (64.3) | 69.3 (82.3) | 68.7 (88.2) | - | 94.3 (16.8) | 87.1 (73.5) | 71.2 (81.7) | - | **100** (0.0) | **100** (0.0) | **100** (0.0) | - |
| **Norm L$_\infty$** | SVHN | CIFAR10 | CIFAR100 | Tiny | SVHN | CIFAR10 | CIFAR100 | Tiny | SVHN | CIFAR10 | CIFAR100 | Tiny |
| PGD$_\infty$ | | | | | | | | | | | | |
| $\varepsilon = 0.03125$ | 93.7 (25.5) | 86.1 (47.1) | 47.4 (95.2) | **100** (0.0) | 95.6 (27.3) | 99.0 (3.5) | 58.6 (88.1) | **100** (0.0) | **100** (0.0) | **100** (0.0) | **100** (0.0) | **100** (0.0) |
| $\varepsilon = 0.0625$ | 98.7 (3.7) | 94.6 (25.2) | 48.8 (93.9) | **100** (0.0) | 99.6 (0.0) | **100** (0.0) | 61.3 (86.3) | **100** (0.0) | **100** (0.0) | **100** (0.0) | **100** (0.0) | **100** (0.0) |
| $\varepsilon = 0.125$ | 99.7 (0.8) | 97.7 (11.2) | 50.3 (92.5) | **100** (0.0) | **100** (0.0) | **100** (0.0) | 68.3 (83.5) | **100** (0.0) | **100** (0.0) | **100** (0.0) | **100** (0.0) | **100** (0.0) |
| $\varepsilon = 0.25$ | 99.6 (2.0) | 99.0 (3.7) | 74.5 (77.0) | 99.9 (0.0) | **100** (0.0) | **100** (0.0) | 79.4 (76.1) | **100** (0.0) | **100** (0.0) | **100** (0.0) | **100** (0.0) | **100** (0.0) |
| $\varepsilon = 0.3125$ | 99.4 (3.2) | 99.1 (3.5) | 77.9 (75.6) | **100** (0.0) | **100** (0.0) | **100** (0.0) | 83.3 (72.5) | **100** (0.0) | **100** (0.0) | **100** (0.0) | **100** (0.0) | **100** (0.0) |
| $\varepsilon = 0.5$ | 98.7 (6.2) | 99.6 (1.1) | 87.2 (56.3) | **100** (0.0) | **100** (0.0) | **100** (0.0) | 90.8 (56.1) | **100** (0.0) | **100** (0.0) | **100** (0.0) | **100** (0.0) | **100** (0.0) |
| BIM | | | | | | | | | | | | |
| $\varepsilon = 0.03125$ | 91.5 (32.6) | 79.7 (63.3) | 47.4 (95.2) | **100** (0.0) | 92.2 (40.4) | 95.8 (22.0) | 58.7 (88.4) | **100** (0.0) | **100** (0.0) | **100** (0.0) | **100** (0.0) | **100** (0.0) |
| $\varepsilon = 0.0625$ | 98.5 (6.3) | 89.2 (42.8) | 48.6 (94.0) | **100** (0.0) | 99.2 (0.6) | **100** (0.0) | 60.7 (86.8) | **100** (0.0) | **100** (0.0) | **100** (0.0) | **100** (0.0) | **100** (0.0) |
| $\varepsilon = 0.125$ | 99.6 (1.6) | 95.8 (21.4) | 49.9 (92.8) | **100** (0.0) | 99.9 (0.0) | **100** (0.0) | 67.1 (83.6) | **100** (0.0) | **100** (0.0) | **100** (0.0) | **100** (0.0) | **100** (0.0) |
| $\varepsilon = 0.25$ | 99.7 (1.2) | 98.6 (4.7) | 74.3 (78.3) | **100** (0.0) | **100** (0.0) | **100** (0.0) | 78.8 (76.5) | **100** (0.0) | **100** (0.0) | **100** (0.0) | **100** (0.0) | **100** (0.0) |
| $\varepsilon = 0.3125$ | 99.4 (3.2) | 99.1 (3.5) | 78.3 (73.8) | **100** (0.0) | **100** (0.0) | **100** (0.0) | 83.4 (72.8) | **100** (0.0) | **100** (0.0) | **100** (0.0) | **100** (0.0) | **100** (0.0) |
| $\varepsilon = 0.5$ | 99.2 (4.3) | 99.7 (1.0) | 85.4 (64.1) | **100** (0.0) | **100** (0.0) | **100** (0.0) | 91.6 (53.6) | **100** (0.0) | **100** (0.0) | **100** (0.0) | **100** (0.0) | **100** (0.0) |
| FGSM | | | | | | | | | | | | |
| $\varepsilon = 0.03125$ | 95.4 (19.4) | 86.6 (49.0) | 48.9 (94.4) | **100** (0.1) | 87.8 (44.0) | 24.9 (99.7) | 40.0 (96.5) | 99.8 (0.6) | **100** (0.0) | **100** (0.0) | **100** (0.0) | **100** (0.0) |
| $\varepsilon = 0.0625$ | 99.2 (3.8) | 97.3 (12.7) | 47.3 (94.2) | **100** (0.0) | 90.7 (31.3) | 81.4 (75.4) | 38.5 (96.9) | **100** (0.0) | **100** (0.0) | **100** (0.0) | **100** (0.0) | **100** (0.0) |
| $\varepsilon = 0.125$ | 99.7 (0.2) | 99.4 (2.9) | 47.2 (92.0) | **100** (0.0) | 92.7 (22.6) | 93.0 (46.3) | 36.0 (97.9) | **100** (0.0) | **100** (0.0) | **100** (0.0) | **100** (0.0) | **100** (0.0) |
| $\varepsilon = 0.25$ | 99.8 (0.0) | 98.4 (3.5) | 82.3 (64.2) | **100** (0.0) | 93.6 (18.0) | 98.8 (5.0) | 67.1 (80.6) | **100** (0.0) | **100** (0.0) | **100** (0.0) | **100** (0.0) | **100** (0.0) |
| $\varepsilon = 0.3125$ | 99.4 (0.0) | 99.1 (1.8) | 86.7 (54.3) | **100** (0.0) | 6.2 (99.5) | 99.2 (3.2) | 69.3 (77.9) | **100** (0.0) | **100** (0.0) | **100** (0.0) | **100** (0.0) | **100** (0.0) |
| $\varepsilon = 0.5$ | 99.9 (0.0) | **100** (0.0) | 93.7 (30.2) | **100** (0.0) | 5.8 (99.5) | 99.6 (1.6) | 73.3 (73.9) | **100** (0.0) | **100** (0.0) | **100** (0.0) | **100** (0.0) | **100** (0.0) |
| CW$_\infty$* | | | | | | | | | | | | |
| $\varepsilon = 0.3125$ | 85.9 (51.3) | 71.5 (80.5) | 78.1 (81.3) | 40.8 (95.4) | 90.0 (32.7) | 79.5 (76.0) | 67.7 (79.7) | 72.0 (72.2) | **100** (0.0) | **100** (0.0) | **100** (0.0) | **100** (0.0) |
| SA | | | | | | | | | | | | |
| $\varepsilon = 0.125$ | 92.3 (31.8) | 93.1 (45.6) | 84.6 (82.0) | 88.1 (84.0) | 93.0 (22.5) | 90.0 (73.4) | 68.9 (78, 4) | 93.6 (47.8) | **100** (0.0) | **100** (0.0) | **100** (0.0) | **100** (0.0) |
| **No norm** | SVHN | CIFAR10 | CIFAR100 | Tiny | SVHN | CIFAR10 | CIFAR100 | Tiny | SVHN | CIFAR10 | CIFAR100 | Tiny |
| No $\varepsilon$ | 99.1 (4.4) | 91.7 (36.6) | 98.4 (4.2) | **100** (0.0) | 92.8 (21.9) | 81.4 (76.2) | 76.1 (61.3) | **100** (0.0) | **100** (0.0) | **100** (0.0) | **100** (0.0) | **100** (0.0) |

# D   Detailed Results on No Knowledge about the attacker

Table 6: Performances on the four considered datasets - SVHN, CIFAR10, CIFAR100, and Tiny ImageNet- of the HAMPER$_{\text{BA}}$ detector together with the results of the state-of-the-art detection method: NSS, on multiple threat scenarios with multiple maximal perturbations $\varepsilon$. The best results among the detectors are shown in **bold**. The results are presented as:   AUROC↑ (FPR $_{\downarrow 95\%}$ %). * stipulates the non-gradient based attacks.

| Norm L$^1$ | NSS | | | | HAMPER$_{\text{BA}}$ | | | |
|---|---|---|---|---|---|---|---|---|
| | SVHN | CIFAR10 | CIFAR100* | Tiny | SVHN | CIFAR10 | CIFAR100* | Tiny |
| PGD$_1$ | | | | | | | | |
| $\varepsilon = 5$ ($\varepsilon^* = 40$) | 48.6 (95.1) | 50.1 (94.3) | 51.6 (94.3) | 44.7 (98.0) | **95.7** (20.4) | **88.4** (47.5) | **99.9** (0.4) | **100** (0.0) |
| $\varepsilon = 10$ ($\varepsilon^* = 500$) | 51.5 (94.6) | 56.7 (90.0) | 53.1 (94.2) | 48.7 (96.6) | **87.6** (51.0) | **90.4** (38.8) | **99.9** (0.3) | **100** (0.0) |
| $\varepsilon = 15$ ($\varepsilon^* = 1000$) | 59.4 (91.7) | 62.5 (85.2) | 58.3 (93.3) | 56.8 (92.5) | **95.1** (23.5) | **89.2** (49.0) | **99.9** (0.2) | **100** (0.0) |
| $\varepsilon = 20$ ($\varepsilon^* = 1500$) | 69.2 (85.1) | 67.7 (80.3) | 66.5 (91.8) | 67.5 (81.3) | **96.0** (21.8) | **98.3** (8.5) | **99.9** (0.2) | **100** (0.0) |
| $\varepsilon = 25$ ($\varepsilon^* = 2000$) | 78.1 (71.7) | 72.0 (74.5) | 75.5 (89.2) | 79.1 (60.6) | **96.4** (19.8) | **98.3** (8.2) | **99.9** (0.2) | **100** (0.0) |
| $\varepsilon = 30$ ($\varepsilon^* = 2500$) | 84.8 (55.5) | 75.9 (68.6) | 83.7 (84.4) | 88.9 (35.6) | **95.7** (27.5) | **98.8** (5.4) | **100** (0.1) | **100** (0.0) |
| $\varepsilon = 40$ ($\varepsilon^* = 5000$) | 92.9 (24.3) | 82.1 (57.1) | 99.4 (0.2) | 98.3 (5.7) | **97.9** (10.1) | **99.3** (2.5) | **100** (0.0) | **100** (0.0) |

| Norm L$_2$ | SVHN | CIFAR10 | CIFAR100* | Tiny | SVHN | CIFAR10 | CIFAR100* | Tiny |
|---|---|---|---|---|---|---|---|---|
| PGD$_2$ | | | | | | | | |
| $\varepsilon = 0.125$ ($\varepsilon^* = 5$) | 49.2 (95.0) | 49.7 (94.2) | 51.5 (94.3) | 87.6 (37.4) | **93.0** (27.7) | **94.6** (26.1) | **99.9** (0.6) | **100** (0.0) |
| $\varepsilon = 0.25$ ($\varepsilon^* = 10$) | 49.6 (94.9) | 55.7 (90.5) | 51.7 (94.0) | 100 (0.0) | **90.4** (37.7) | **89.6** (45.1) | **99.9** (0.3) | **100** (0.0) |
| $\varepsilon = 0.3125$ ($\varepsilon^* = 15$) | 52.5 (94.1) | 59.0 (88.2) | 52.8 (94.2) | 100 (0.0) | **91.9** (34.0) | 79.9 (69.9) | **99.8** (0.8) | **100** (0.0) |
| $\varepsilon = 0.5$ ($\varepsilon^* = 20$) | 66.4 (87.4) | 67.5 (79.9) | 54.4 (94.4) | 100 (0.0) | **94.0** (30.7) | **94.7** (30.6) | **99.9** (0.2) | **100** (0.0) |
| $\varepsilon = 1$ ($\varepsilon^* = 30$) | 92.1 (29.6) | 83.1 (54.4) | 59.1 (93.6) | 100 (0.0) | **98.8** (6.5) | **99.4** (3.3) | **100** (0.1) | **100** (0.0) |
| $\varepsilon = 1.5$ ($\varepsilon^* = 40$) | 98.0 (5.9) | 91.7 (32.8) | 67.2 (92.4) | 100 (0.0) | **98.9** (4.4) | **100** (0.0) | **99.9** (0.2) | **100** (0.0) |
| $\varepsilon = 2$ ($\varepsilon^* = 50$) | 99.4 (1.6) | 96.2 (16.1) | 77.5 (88.4) | 100 (0.0) | **99.5** (2.4) | **100** (0.0) | **100** (0.1) | **100** (0.0) |
| DeepFool* | | | | | | | | |
| No $\varepsilon$ | 58.2 (93.4) | 55.9 (92.3) | 73.0 (72.6) | - | **90.8** (29.0) | **79.7** (62.5) | **100** (0.0) | - |
| CW$_2$* | | | | | | | | |
| $\varepsilon = 0.01$ | 61.8 (92.0) | 56.0 (91.2) | 64.6 (90.0) | | **92.7** (30.9) | **89.2** (44.6) | **99.9** (0.1) | - |
| HOP | | | | | | | | |
| $\varepsilon = 0.1$ | 87.6 (64.0) | 65.4 (87.9) | 73.2 (87.9) | - | **93.8** (22.5) | **95.6** (19.6) | **99.8** (0.5) | - |

| Norm L$_\infty$ | SVHN | CIFAR10 | CIFAR100 | Tiny | SVHN | CIFAR10 | CIFAR100 | Tiny |
|---|---|---|---|---|---|---|---|---|
| PGD$_\infty$ | | | | | | | | |
| $\varepsilon = 0.03125$ | **99.3** (1.7) | 91.3 (34.2) | 53.1 (93.2) | **100** (0.0) | 97.4 (13.8) | **99.4** (2.3) | **99.9** (0.4) | **100** (0.0) |
| $\varepsilon = 0.0625$ | **99.9** (0.2) | 99.0 (4.4) | 55.9 (91.8) | **100** (0.0) | 99.5 (2.0) | **99.3** (3.6) | **100** (0.1) | **100** (0.0) |
| $\varepsilon = 0.125$ | **99.9** (0.2) | **99.9** (0.3) | 61.5 (89.9) | **100** (0.0) | **99.9** (0.3) | 99.8 (0.7) | **99.9** (0.4) | **100** (0.0) |
| $\varepsilon = 0.25$ | **99.9** (0.2) | 99.9 (0.1) | 71.3 (84.9) | **100** (0.0) | 99.7 (1.7) | **100** (0.0) | **99.9** (0.4) | **100** (0.0) |
| $\varepsilon = 0.3125$ | 99.9 (0.2) | **99.9** (0.1) | 75.4 (81.3) | **100** (0.0) | **100** (0.1) | 99.8 (0.8) | **99.8** (0.8) | **100** (0.0) |
| $\varepsilon = 0.5$ | 99.9 (0.2) | **99.9** (0.1) | 84.7 (67.6) | **100** (0.0) | **100** (0.2) | **100** (0.1) | **99.8** (0.5) | **100** (0.0) |
| BIM | | | | | | | | |
| $\varepsilon = 0.03125$ | **99.0** (3.0) | 89.3 (41.4) | 53.0 (93.3) | **100** (0.0) | 96.7 (21.2) | **96.8** (14.4) | **99.9** (0.3) | **100** (0.0) |
| $\varepsilon = 0.0625$ | **99.8** (0.3) | 97.9 (9.2) | 55.8 (91.9) | **100** (0.0) | 99.2 (3.8) | **99.8** (0.8) | **99.9** (0.2) | **100** (0.0) |
| $\varepsilon = 0.125$ | **99.9** (0.2) | 99.7 (0.9) | 61.4 (90.3) | **100** (0.0) | **99.9** (0.7) | **99.8** (1.1) | **99.9** (0.4) | **100** (0.0) |
| $\varepsilon = 0.25$ | 99.9 (0.2) | **99.9** (0.1) | 71.2 (85.1) | **100** (0.0) | **100** (0.0) | **99.9** (0.4) | **99.9** (0.4) | **100** (0.0) |
| $\varepsilon = 0.3125$ | **99.9** (0.2) | **99.9** (0.1) | 75.3 (81.0) | **100** (0.0) | **99.9** (0.6) | 99.8 (1.0) | **99.9** (0.4) | **100** (0.0) |
| $\varepsilon = 0.5$ | **99.9** (0.2) | **99.9** (0.1) | 84.9 (67.7) | **100** (0.0) | **99.9** (0.2) | 99.7 (1.4) | **99.9** (0.1) | **100** (0.0) |
| FGSM | | | | | | | | |
| $\varepsilon = 0.03125$ | **99.6** (0.9) | 93.6 (28.2) | 53.1 (93.6) | **100** (0.0) | 94.3 (26.2) | **96.9** (15.7) | **99.8** (0.6) | **100** (0.0) |
| $\varepsilon = 0.0625$ | 98.7 (0.2) | **99.6** (1.5) | 57.1 (90.9) | **100** (0.0) | **95.5** (20.3) | 94.6 (27.0) | **99.9** (0.2) | **100** (0.0) |
| $\varepsilon = 0.125$ | 82.0 (100) | **99.9** (0.1) | 67.2 (87.0) | **100** (0.0) | **98.6** (6.5) | 99.7 (1.1) | **99.9** (0.1) | **100** (0.0) |
| $\varepsilon = 0.25$ | 70.5 (100) | **99.9** (0.1) | 82.1 (74.0) | **100** (0.0) | **99.4** (3.2) | 99.9 (0.5) | **99.9** (0.4) | **100** (0.0) |
| $\varepsilon = 0.3125$ | 76.1 (100) | **99.9** (0.1) | 86.9 (64.5) | **100** (0.0) | **98.5** (7.2) | **100** (0.1) | **99.9** (0.4) | **100** (0.0) |
| $\varepsilon = 0.5$ | 88.7 (97.6) | **99.9** (0.1) | 94.1 (36.9) | **100** (0.0) | **99.8** (0.7) | **100** (0.0) | **99.9** (0.2) | **100** (0.0) |
| CW$_\infty$* | | | | | | | | |
| $\varepsilon = 0.3125$ | 67.9 (90.9) | 64.6 (89.9) | 70.0 (85.3) | **100** (0.0) | **90.5** (33.0) | **90.5** (40.9) | **99.9** (0.1) | **100** (0.0) |
| SA | | | | | | | | |
| $\varepsilon = 0.125$ | 91.3 (82.3) | 11.6 (99.9) | 67.4 (89.3) | 90.2 (74.5) | **99.0** (4.9) | **97.9** (12.7) | **99.9** (0.2) | **100** (0.0) |

| No norm | SVHN | CIFAR10 | CIFAR100 | Tiny | SVHN | CIFAR10 | CIFAR100 | Tiny |
|---|---|---|---|---|---|---|---|---|
| No $\varepsilon$ | **99.8** (6.4) | **93.8** (20.2) | 92.9 (24.7) | **100** (0.0) | 98.5 (0.4) | 80.3 (57.1) | **100** (0.0) | **100** (0.0) |

# E    Comparison to Additional State-Of-The-Art Methods

We compared our proposed detector with two additional methods: Mahalanobis, introduced by Lee et al. (2018), which works under the framework of Attack-Aware detectors, and JTLA, introduced by Raghuram et al. (2021), which is an unsupervised method. We reported the detailed results in Tab. 7 and Tab. 8, and the averaged ones in Tab. 9.

Table 7: Performances on the four considered datasets - SVHN, CIFAR10, CIFAR100 and Tiny ImageNet- of the HAMPER$_{AA}$ detector together with the results of the state-of-the-art detection methods: Mahalanobis, on multiple threat scenarios with multiple maximal perturbations $\varepsilon$. The best results among the detectors are shown in **bold**. The results are presented as:    AUROC$\uparrow$ (FPR $\downarrow_{95\%}$ %). $^*$ stipulates the non-gradient based attacks.

| Norm $L_1$ | Mahalanobis | | | | HAMPER$_{AA}$ | | | |
|---|---|---|---|---|---|---|---|---|
| | SVHN | CIFAR10 | CIFAR100 | Tiny | SVHN | CIFAR10 | CIFAR100 | Tiny |
| PGD$_1$ | | | | | | | | |
| $\varepsilon = 5$ ($\varepsilon^\star = 40$) | 81.3 (80.5) | 80.2 (83.8) | 75.2 (89.2) | 95.6 (19.5) | **100** (0.0) | **100** (0.0) | **100** (0.0) | **100** (0.0) |
| $\varepsilon = 10$ ($\varepsilon^\star = 500$) | 85.4 (73.0) | 84.9 (71.5) | 79.3 (83.3) | 98.7 (4.7) | **100** (0.0) | **100** (0.0) | **100** (0.0) | **100** (0.0) |
| $\varepsilon = 15$ ($\varepsilon^\star = 1000$) | 89.2 (58.3) | 88.6 (58.4) | 80.7 (79.4) | 99.0 (3.8) | **100** (0.0) | **100** (0.0) | **100** (0.0) | **100** (0.0) |
| $\varepsilon = 20$ ($\varepsilon^\star = 1500$) | 92.9 (39.3) | 93.4 (35.3) | 85.7 (65.4) | 99.8 (0.1) | **100** (0.0) | **100** (0.0) | **100** (0.0) | **100** (0.0) |
| $\varepsilon = 25$ ($\varepsilon^\star = 2000$) | 95.6 (24.7) | 96.8 (18.4) | 91.0 (46.0) | 99.9 (0.1) | **100** (0.0) | **100** (0.0) | **100** (0.0) | **100** (0.0) |
| $\varepsilon = 30$ ($\varepsilon^\star = 2500$) | 97.3 (25.3) | 98.5 (8.3) | 95.2 (26.2) | **100** (0.0) | **100** (0.0) | **100** (0.0) | **100** (0.0) | **100** (0.0) |
| $\varepsilon = 40$ ($\varepsilon^\star = 5000$) | 98.9 (5.7) | 99.7 (1.8) | 99.9 (0.3) | **100** (0.0) | **100** (0.0) | **100** (0.0) | **100** (0.0) | **100** (0.0) |
| **Norm $L_2$** | SVHN | CIFAR10 | CIFAR100 | Tiny | SVHN | CIFAR10 | CIFAR100 | Tiny |
| PGD$_2$ | | | | | | | | |
| $\varepsilon = 0.125$ ($\varepsilon^\star = 5$) | 77.9 (81.8) | 78.1 (84.2) | 77.5 (86.3) | 99.9 (0.1) | **100** (0.0) | **100** (0.0) | **100** (0.0) | **100** (0.0) |
| $\varepsilon = 0.25$ ($\varepsilon^\star = 10$) | 84.0 (74.7) | 84.2 (71.9) | 79.0 (83.6) | **100** (0.1) | **100** (0.0) | **100** (0.0) | **100** (0.0) | **100** (0.0) |
| $\varepsilon = 0.3125$ ($\varepsilon^\star = 15$) | 86.0 (69.2) | 86.4 (65.5) | 79.8 (82.7) | **100** (0.0) | **100** (0.0) | **100** (0.0) | **100** (0.0) | **100** (0.0) |
| $\varepsilon = 0.5$ ($\varepsilon^\star = 20$) | 91.8 (44.3) | 93.2 (35.6) | 80.3 (82.5) | **100** (0.0) | **100** (0.0) | **100** (0.0) | **100** (0.0) | **100** (0.0) |
| $\varepsilon = 1$ ($\varepsilon^\star = 30$) | 98.6 (7.7) | 99.5 (2.4) | 82.5 (75.6) | **100** (0.0) | **100** (0.0) | **100** (0.0) | **100** (0.0) | **100** (0.0) |
| $\varepsilon = 1.5$ ($\varepsilon^\star = 40$) | 99.7 (1.3) | 99.8 (0.8) | 88.0 (58.9) | **100** (0.0) | **100** (0.0) | **100** (0.0) | **100** (0.0) | **100** (0.0) |
| $\varepsilon = 2$ ($\varepsilon^\star = 50$) | 99.9 (0.4) | 99.8 (0.8) | 93.3 (36.8) | **100** (0.0) | **100** (0.0) | **100** (0.0) | **100** (0.0) | **100** (0.0) |
| DeepFool$^*$ | | | | | | | | |
| No $\varepsilon$ | 81.0 (74.1) | 75.8 (79.2) | 89.8 (50.2) | - | **100** (0.0) | **100** (0.0) | **100** (0.0) | - |
| CW$_2$$^*$ | | | | | | | | |
| $\varepsilon = 0.01$ | 74.5 (79.5) | 66.4 (86.6) | 58.6 (87.9) | - | **100** (0.0) | **100** (0.0) | **100** (0.0) | - |
| HOP | | | | | | | | |
| $\varepsilon = 0.1$ | 79.9 (74.4) | 76.7 (77.7) | 56.9 (90.2) | - | **100** (0.0) | **100** (0.0) | **100** (0.0) | - |
| **Norm $L_\infty$** | SVHN | CIFAR10 | CIFAR100 | Tiny | SVHN | CIFAR10 | CIFAR100 | Tiny |
| PGD$_\infty$ | | | | | | | | |
| $\varepsilon = 0.03125$ | 98.2 (9.3) | 99.3 (2.8) | 52.9 (94.2) | **100** (0.0) | **100** (0.0) | **100** (0.0) | **100** (0.0) | **100** (0.0) |
| $\varepsilon = 0.0625$ | **100** (0.1) | **100** (0.0) | 54.7 (88.3) | **100** (0.0) | **100** (0.0) | **100** (0.0) | **100** (0.0) | **100** (0.0) |
| $\varepsilon = 0.125$ | **100** (0.0) | **100** (0.0) | 70.7 (72.7) | **100** (0.0) | **100** (0.0) | **100** (0.0) | **100** (0.0) | **100** (0.0) |
| $\varepsilon = 0.25$ | **100** (0.0) | **100** (0.0) | 85.7 (46.5) | **100** (0.0) | **100** (0.0) | **100** (0.0) | **100** (0.0) | **100** (0.0) |
| $\varepsilon = 0.3125$ | **100** (0.0) | **100** (0.0) | 89.7 (62.4) | **100** (0.0) | **100** (0.0) | **100** (0.0) | **100** (0.0) | **100** (0.0) |
| $\varepsilon = 0.5$ | **100** (0.0) | **100** (0.0) | 95.7 (18.1) | **100** (0.0) | **100** (0.0) | **100** (0.0) | **100** (0.0) | **100** (0.0) |
| BIM | | | | | | | | |
| $\varepsilon = 0.03125$ | 96.6 (16.9) | 97.4 (12.8) | 52.5 (94.6) | **100** (0.0) | **100** (0.0) | **100** (0.0) | **100** (0.0) | **100** (0.0) |
| $\varepsilon = 0.0625$ | 99.9 (0.3) | **100** (0.1) | 58.3 (89.3) | **100** (0.0) | **100** (0.0) | **100** (0.0) | **100** (0.0) | **100** (0.0) |
| $\varepsilon = 0.125$ | **100** (0.0) | **100** (0.0) | 69.8 (75.1) | **100** (0.0) | **100** (0.0) | **100** (0.0) | **100** (0.0) | **100** (0.0) |
| $\varepsilon = 0.25$ | **100** (0.0) | **100** (0.0) | 85.3 (47.4) | **100** (0.0) | **100** (0.0) | **100** (0.0) | **100** (0.0) | **100** (0.0) |
| $\varepsilon = 0.3125$ | **100** (0.0) | **100** (0.0) | 89.7 (37.0) | **100** (0.0) | **100** (0.0) | **100** (0.0) | **100** (0.0) | **100** (0.0) |
| $\varepsilon = 0.5$ | **100** (0.0) | **100** (0.0) | 96.1 (16.4) | **100** (0.0) | **100** (0.0) | **100** (0.0) | **100** (0.0) | **100** (0.0) |
| FGSM | | | | | | | | |
| $\varepsilon = 0.03125$ | 94.8 (27.9) | **100** (0.0) | 52.4 (94.4) | **100** (0.0) | **100** (0.0) | **100** (0.0) | **100** (0.0) | **100** (0.0) |
| $\varepsilon = 0.0625$ | 99.4 (1.3) | 90.8 (46.3) | 55.2 (92.8) | **100** (0.0) | **100** (0.0) | **100** (0.0) | **100** (0.0) | **100** (0.0) |
| $\varepsilon = 0.125$ | 99.9 (0.0) | 96.5 (17.8) | 59.0 (91.2) | **100** (0.0) | **100** (0.0) | **100** (0.0) | **100** (0.0) | **100** (0.0) |
| $\varepsilon = 0.25$ | **100** (0.0) | 99.5 (1.9) | 64.2 (89.0) | **100** (0.0) | **100** (0.0) | **100** (0.0) | **100** (0.0) | **100** (0.0) |
| $\varepsilon = 0.3125$ | **100** (0.0) | 99.8 (0.8) | 66.2 (91.6) | **100** (0.0) | **100** (0.0) | **100** (0.0) | **100** (0.0) | **100** (0.0) |
| $\varepsilon = 0.5$ | **100** (0.0) | **100** (0.0) | 73.2 (88.6) | **100** (0.0) | **100** (0.0) | **100** (0.0) | **100** (0.0) | **100** (0.0) |
| CW$_\infty$$^*$ | | | | | | | | |
| $\varepsilon = 0.3125$ | 90.6 (52.9) | 85.3 (74.2) | 77.5 (83.4) | **100** (0.0) | **100** (0.0) | **100** (0.0) | **100** (0.0) | **100** (0.0) |
| SA | | | | | | | | |
| $\varepsilon = 0.125$ | 81.0 (53.3) | 96.8 (9.6) | 79.2 (63.6) | 95.6 (17.0) | **100** (0.0) | **100** (0.0) | **100** (0.0) | **100** (0.0) |
| **No norm** | SVHN | CIFAR10 | CIFAR100 | Tiny | SVHN | CIFAR10 | CIFAR100 | Tiny |
| No $\varepsilon$ | **100** (0.0) | 81.5 (50.5) | 93.0 (32.9) | **100** (0.0) | **100** (0.0) | **100** (0.0) | **100** (0.0) | **100** (0.0) |

Table 8: Performances on the four considered datasets - SVHN, CIFAR10, CIFAR100 and Tiny ImageNet- of the HAMPER_BA detector together with the results of the state-of-the-art detection methods: JTLA, on multiple threat scenarios with multiple maximal perturbations $\varepsilon$. The best results among the detectors are shown in **bold**. The results are presented as: AUROC↑ (FPR $\downarrow_{95\%}$ %). * stipulates the non-gradient based attacks.

| Norm $L_1$ | JTLA | | | | HAMPER_BA | | | |
|---|---|---|---|---|---|---|---|---|
| | SVHN | CIFAR10 | CIFAR100 | Tiny | SVHN | CIFAR10 | CIFAR100 | Tiny |
| PGD₁ | | | | | | | | |
| $\varepsilon = 5$ ($\varepsilon^\star = 40$) | 82.1 (55.0) | 73.0 (73.2) | 57.7 (89.2) | 50.2 (94.2) | **95.7** (20.4) | **88.4** (47.5) | **99.9** (0.4) | **100** (0.0) |
| $\varepsilon = 10$ ($\varepsilon^\star = 500$) | 78.9 (59.9) | 65.2 (83.2) | 59.0 (89.4) | 47.1 (95.7) | **87.6** (51.0) | **90.4** (38.8) | **99.9** (0.3) | **100** (0.0) |
| $\varepsilon = 15$ ($\varepsilon^\star = 1000$) | 77.5 (62.0) | 61.5 (86.5) | 59.2 (88.7) | 45.3 (96.5) | **95.1** (23.5) | **89.2** (49.0) | **99.9** (0.2) | **100** (0.0) |
| $\varepsilon = 20$ ($\varepsilon^\star = 1500$) | 76.3 (64.0) | 59.4 (88.3) | 57.2 (88.6) | 43.7 (96.8) | **96.0** (21.8) | **98.3** (8.5) | **99.9** (0.2) | **100** (0.0) |
| $\varepsilon = 25$ ($\varepsilon^\star = 2000$) | 75.4 (65.3) | 57.7 (89.5) | 55.2 (89.1) | 41.8 (96.7) | **96.4** (19.8) | **98.3** (8.2) | **99.9** (0.2) | **100** (0.0) |
| $\varepsilon = 30$ ($\varepsilon^\star = 2500$) | 74.4 (66.5) | 56.6 (90.2) | 54.1 (88.8) | 43.0 (96.7) | **95.7** (27.5) | **98.8** (5.4) | **100** (0.1) | **100** (0.0) |
| $\varepsilon = 40$ ($\varepsilon^\star = 5000$) | 72.6 (68.8) | 54.9 (90.9) | 42.0 (90.1) | 43.0 (96.9) | **97.9** (10.1) | **99.3** (2.5) | **100** (0.0) | **100** (0.0) |
| **Norm $L_2$** | SVHN | CIFAR10 | CIFAR100 | Tiny | SVHN | CIFAR10 | CIFAR100 | Tiny |
| PGD₂ | | | | | | | | |
| $\varepsilon = 0.125$ ($\varepsilon^\star = 5$) | 83.8 (52.3) | 74.4 (71.7) | 58.3 (88.7) | 40.0 (97.2) | **93.0** (27.7) | **94.6** (26.1) | **99.9** (0.6) | **100** (0.0) |
| $\varepsilon = 0.25$ ($\varepsilon^\star = 10$) | 80.0 (58.2) | 66.0 (82.5) | 58.8 (88.7) | 39.5 (97.4) | **90.4** (37.7) | **89.6** (45.1) | **99.9** (0.3) | **100** (0.0) |
| $\varepsilon = 0.3125$ ($\varepsilon^\star = 15$) | 78.9 (59.9) | 63.6 (84.4) | 59.3 (88.6) | 39.8 (97.4) | **91.9** (34.0) | **79.9** (69.9) | **99.8** (0.8) | **100** (0.0) |
| $\varepsilon = 0.5$ ($\varepsilon^\star = 20$) | 76.8 (63.4) | 59.3 (88.5) | 59.5 (89.1) | 39.5 (97.4) | **94.0** (30.7) | **94.7** (30.6) | **99.9** (0.2) | **100** (0.0) |
| $\varepsilon = 1$ ($\varepsilon^\star = 30$) | 72.7 (68.6) | 54.9 (91.1) | 58.3 (87.6) | 39.4 (97.4) | **98.8** (6.5) | **99.4** (3.3) | **100** (0.1) | **100** (0.0) |
| $\varepsilon = 1.5$ ($\varepsilon^\star = 40$) | 69.3 (72.8) | 53.5 (91.2) | 55.5 (87.5) | 39.8 (97.4) | **98.9** (4.4) | **100** (0.0) | **99.9** (0.2) | **100** (0.0) |
| $\varepsilon = 2$ ($\varepsilon^\star = 50$) | 66.9 (75.3) | 53.1 (91.3) | 52.1 (88.5) | 39.5 (97.4) | **99.5** (2.4) | **100** (0.0) | **100** (0.1) | **100** (0.0) |
| DeepFool* | | | | | | | | |
| No $\varepsilon$ | **95.8** (11.6) | **94.6** (33.7) | 63.9 (77.9) | - | 90.8 (29.0) | 79.7 (62.5) | **100** (0.0) | - |
| CW₂* | | | | | | | | |
| $\varepsilon = 0.01$ | 77.6 (60.9) | 66.3 (78.4) | 58.4 (85.5) | - | **92.7** (30.9) | **89.2** (44.6) | **99.9** (0.1) | - |
| HOP | | | | | | | | |
| $\varepsilon = 0.1$ | **94.5** (18.1) | 91.1 (34.4) | 59.9 (83.6) | - | 93.8 (22.5) | **95.6** (19.6) | **99.8** (0.5) | - |
| **Norm $L_\infty$** | SVHN | CIFAR10 | CIFAR100 | Tiny | SVHN | CIFAR10 | CIFAR100 | Tiny |
| PGD∞ | | | | | | | | |
| $\varepsilon = 0.03125$ | 72.9 (68.9) | 55.9 (91.4) | 56.4 (90.4) | 31.4 (91.6) | **97.4** (13.8) | **99.4** (2.3) | **99.9** (0.4) | **100** (0.0) |
| $\varepsilon = 0.0625$ | 66.5 (75.9) | 54.7 (91.3) | 54.8 (91.1) | 18.9 (92.0) | **99.5** (2.0) | **99.3** (3.6) | **100** (0.1) | **100** (0.0) |
| $\varepsilon = 0.125$ | 61.3 (78.5) | 55.3 (90.9) | 52.3 (91.8) | 19.9 (91.9) | **99.9** (0.3) | **99.8** (0.7) | **99.9** (0.4) | **100** (0.0) |
| $\varepsilon = 0.25$ | 59.0 (79.8) | 56.1 (90.7) | 49.9 (92.9) | 20.5 (91.9) | **99.7** (1.7) | **100** (0.0) | **99.9** (0.4) | **100** (0.0) |
| $\varepsilon = 0.3125$ | 58.5 (80.0) | 56.2 (90.7) | 50.0 (93.0) | 20.6 (91.9) | **100** (0.1) | **99.8** (0.8) | **99.8** (0.8) | **100** (0.0) |
| $\varepsilon = 0.5$ | 58.2 (80.1) | 56.1 (90.8) | 49.0 (93.4) | 20.6 (91.9) | **100** (0.2) | **100** (0.1) | **99.8** (0.5) | **100** (0.0) |
| BIM | | | | | | | | |
| $\varepsilon = 0.03125$ | 73.3 (69.3) | 56.5 (91.0) | 56.8 (90.1) | 53.3 (78.7) | **96.7** (21.2) | **96.8** (14.4) | **99.9** (0.3) | **100** (0.0) |
| $\varepsilon = 0.0625$ | 67.0 (76.3) | 54.7 (91.4) | 55.0 (91.3) | 40.5 (90.9) | **99.2** (3.8) | **99.8** (0.8) | **99.9** (0.2) | **100** (0.0) |
| $\varepsilon = 0.125$ | 62.1 (79.0) | 55.2 (91.0) | 52.5 (91.7) | 28.1 (91.7) | **99.9** (0.7) | **99.8** (1.1) | **99.9** (0.4) | **100** (0.0) |
| $\varepsilon = 0.25$ | 59.2 (79.9) | 55.8 (90.9) | 50.4 (92.7) | 20.9 (91.9) | **100** (0.0) | **99.9** (0.4) | **99.9** (0.4) | **100** (0.0) |
| $\varepsilon = 0.3125$ | 58.6 (80.0) | 56.1 (90.7) | 49.5 (92.5) | 20.5 (91.9) | **99.9** (0.6) | **99.8** (1.0) | **99.9** (0.4) | **100** (0.0) |
| $\varepsilon = 0.5$ | 57.5 (80.1) | 56.7 (90.2) | 49.4 (93.7) | 21.6 (91.9) | **99.9** (0.2) | **99.7** (1.4) | **99.9** (0.1) | **100** (0.0) |
| FGSM | | | | | | | | |
| $\varepsilon = 0.03125$ | 83.9 (57.0) | 70.3 (89.3) | 57.7 (89.2) | 54.5 (76.8) | **94.3** (26.2) | **96.9** (15.7) | **99.8** (0.6) | **100** (0.0) |
| $\varepsilon = 0.0625$ | 85.4 (52.5) | 67.9 (95.7) | 59.0 (87.9) | 54.5 (76.8) | **95.5** (20.3) | **94.6** (27.0) | **99.9** (0.2) | **100** (0.0) |
| $\varepsilon = 0.125$ | 87.7 (45.6) | 65.7 (96.0) | 59.7 (86.4) | 54.5 (76.8) | **98.6** (6.5) | **99.7** (1.1) | **99.9** (0.1) | **100** (0.0) |
| $\varepsilon = 0.25$ | 88.9 (42.4) | 65.1 (84.4) | 57.1 (84.9) | 54.6 (76.7) | **99.4** (3.2) | **99.9** (0.5) | **99.9** (0.4) | **100** (0.0) |
| $\varepsilon = 0.3125$ | 89.0 (41.6) | 65.9 (72.7) | 57.3 (83.9) | 54.5 (76.8) | **98.5** (7.2) | **100** (0.1) | **99.9** (0.4) | **100** (0.0) |
| $\varepsilon = 0.5$ | 88.8 (41.8) | 74.2 (55.1) | 56.8 (83.6) | 54.5 (76.8) | **99.8** (0.7) | **100** (0.0) | **99.9** (0.2) | **100** (0.0) |
| CW∞* | | | | | | | | |
| $\varepsilon = 0.3125$ | 89.7 (43.1) | 83.1 (54.4) | 60.5 (82.3) | 51.1 (85.0) | **90.5** (33.0) | **90.5** (40.9) | **99.9** (0.1) | **100** (0.0) |
| SA | | | | | | | | |
| $\varepsilon = 0.125$ | 90.4 (41.2) | 88.7 (37.8) | 61.8 (80.5) | 54.9 (77.8) | **99.0** (4.9) | **97.9** (12.7) | **99.9** (0.2) | **100** (0.0) |
| **No norm** | SVHN | CIFAR10 | CIFAR100 | Tiny | SVHN | CIFAR10 | CIFAR100 | Tiny |
| No $\varepsilon$ | 84.7 (51.7) | **85.0** (52.6) | 60.2 (76.5) | 54.5 (76.8) | **98.5** (0.4) | 80.3 (57.1) | **100** (0.0) | **100** (0.0) |

# F    Additional Attacks

To further assess the strength of our detector, we tested all considered methods on two additional attacks on CIFAR10: a targeted version of $CW_2$ (reported as $CW_{2,targeted}$) and a version of $CW_2$ with higher confidence (kappa=30), both of which don't clip the adversarial examples according to a specific $L_p$-norm. The results are reported in Tab. 10. It is clear from the results that our proposed framework outperforms the other state-of-the-art methods on those specific attacks.

Table 9: Attack-aware performances on the three considered datasets - SVHN, CIFAR10 and CIFAR100 - of `HAMPER`AA and `HAMPER`BA detector together with the results of the SOTA detection methods: `Mahalanobis` and `JTLA` averaged over the $L_p$-norm constraint. The best results among the detectors are shown in **bold**. The results are presented as AUROC↑ (FPR $\downarrow_{95\%}$ %) and in terms of mean ($\mu$) and standard deviation ($\sigma$).

| | | Attack-Aware | | | | | | | | Blind-to-Attack | | | | | | | |
| | | Mahalanobis | | | | HAMPER$_{AA}$ | | | | JTLA | | | | HAMPER$_{BA}$ | | | |
| | | SVHN | CIFAR10 | CIFAR100 | Tiny | SVHN | CIFAR10 | CIFAR100 | Tiny | SVHN | CIFAR10 | CIFAR100 | Tiny | SVHN | CIFAR10 | CIFAR100 | Tiny |
|---|---|---|---|---|---|---|---|---|---|---|---|---|---|---|---|---|---|
| **Norm L$_1$** | $\mu$ | 91.5 (43.8) | 91.7 (39.6) | 86.7 (55.7) | 99.0 (4.0) | **100** (0.0) | **100** (0.0) | **100** (0.0) | **100** (0.0) | 76.7 (63.1) | 61.1 (86.0) | 54.9 (89.1) | 44.9 (96.2) | 94.9 (24.9) | 94.7 (22.8) | 99.9 (0.2) | **100** (0) |
| | $\sigma$ | 6.0 (25.6) | 6.8 (29.7) | 8.4 (30.6) | 1.5 (6.6) | 0.0 (0.0) | 0.0 (0.0) | 0.0 (0.0) | 0.0 (0.0) | 2.9 (4.3) | 5.8 (5.7) | 5.5 (0.5) | 2.7 (0.9) | 3.1 (11.7) | 4.7 (19.6) | 0.0 (0.1) | 0.0 (0.0) |
| **Norm L$_2$** | $\mu$ | 87.6 (50.7) | 86.0 (50.5) | 78.6 (73.5) | **100** (0.0) | **100** (0.0) | **100** (0.0) | **100** (0.0) | **100** (0.0) | 79.6 (54.1) | 67.7 (74.7) | 58.4 (86.6) | 39.6 (97.3) | 94.2 (22.8) | 92.3 (30.2) | 99.9 (0.3) | **100** (0.0) |
| | $\sigma$ | 9.5 (32.7) | 11.2 (34.9) | 11.5 (17.4) | 0.0 (0.0) | 0.0 (0.0) | 0.0 (0.0) | 0.0 (0.0) | 0.0 (0.0) | 9.1 (20.7) | 14.1 (31.2) | 2.9 (3.3) | 0.2 (0.1) | 3.4 (12.6) | 7.2 (23.9) | 0.1 (0.2) | 0.0 (0.0) |
| **Norm L$_\infty$** | $\mu$ | 98.0 (8.6) | 98.3 (8.3) | 71.4 (71.8) | 99.8 (0.9) | **100** (0.0) | **100** (0.0) | **100** (0.0) | **100** (0.0) | 72.9 (64.6) | 62.5 (83.8) | 54.8 (89.2) | 37.5 (85.6) | 98.4 (7.3) | 98.7 (6.3) | 99.9 (0.3) | **100** (0.0) |
| | $\sigma$ | 4.5 (16.5) | 3.7 (18.5) | 14.8 (25.1) | 1.0 (3.7) | 0.0 (0.0) | 0.0 (0.0) | 0.0 (0.0) | 0.0 (0.0) | 13.1 (16.1) | 9.8 (15.6) | 4.0 (4.0) | 15.7 (7.1) | 2.4 (9.8) | 2.3 (10.7) | 0.0 (0.2) | 0.0 (0.0) |
| **No Norm** | $\mu$ | **100** (0.0) | 81.5 (50.5) | 93.0 (32.9) | **100** (0.0) | **100** (0.0) | **100** (0.0) | **100** (0.0) | **100** (0.0) | 84.7 (51.7) | **85.0** (52.6) | 60.2 (76.5) | 54.5 (76.8) | 98.5 (6.4) | 80.3 (57.1) | **100** (0.0) | **100** (0.0) |
| | $\sigma$ | 0.0 (0) | 0.0 (0.0) | 0.0 (0.0) | 0.0 (0.0) | 0.0 (0.0) | 0.0 (0.0) | 0.0 (0.0) | 0.0 (0.0) | 0.0 (0.0) | 0.0 (0.0) | 0.0 (0.0) | 0.0 (0.0) | 0.0 (0.0) | 0.0 (0.0) | 0.0 (0.0) | 0.0 (0.0) |
| **Average** | $\mu$ | 94.3 (25.9) | 93.4 (26.3) | 76.7 (68.3) | 99.7 (1.3) | **100** (0.0) | **100** (0.0) | **100** (0.0) | **100** (0.0) | 75.7 (61.2) | 64.2 (83.8) | 55.9 (88.1) | 39.9 (89.8) | 96.6 (14.6) | 95.8 (17.0) | 99.9 (0.3) | **100** (0.0) |
| | $\sigma$ | 7.8 (30.7) | 8.9 (32.2) | 14.3 (25.7) | 1.0 (4.3) | 0.0 (0.0) | 0.0 (0.0) | 0.0 (0.0) | 0.0 (0.0) | 11.1 (16.6) | 11.2 (17.1) | 4.4 (4.0) | 12.5 (8.0) | 3.4 (13.6) | 5.9 (20.9) | 0.1 (0.2) | 0.0 (0.0) |
| **Global** | $\mu$ | 91.0 (30.5) | | | | **100.0** (0.0) | | | | 58.9 (80.7) | | | | **98.1** (8.0) | | | |
| | $\sigma$ | 8.0 (23.2) | | | | 0.0 (0.0) | | | | 9.8 (11.4) | | | | 2.35 (8.6) | | | |

Table 10: Results on CIFAR10 for `LID`, `KD-BU`, `NSS`, `HAMPER`AA, and `HAMPER`AA on CW$_2$ with high confidence (i.e., CW$_{2,\kappa=30}$) and CW$_2$ targeted (i.e., CW$_{2,\text{targeted}}$).

| Attack | Attack-Aware | | | Blind-to-Attack | |
| | LID | KD-BU | HAMPER$_{AA}$ | NSS | HAMPER$_{BA}$ |
|---|---|---|---|---|---|
| CW$_{2,\kappa=30}$ | 54.0 (92.6) | 50.1 (94.8) | **100** (0.0) | 62.0 (88.6) | **96.9** (15.0) |
| CW$_{2,\text{targeted}}$ | 37.3 (98.2) | 52.6 (93.4) | **100** (0.0) | 46.9 (95.1) | **97.8** (12.6) |

# G  Sanity Check

Recently, (Tramer, 2022) presented a way to compare robust and detection performances between methods. They claim that, for a detector that can achieve $1 - R_{adv}$ performances for a given $\varepsilon$, it is possible to create a robust classifier with the same $1 - R_{adv}$, but for $\varepsilon^* = \frac{\varepsilon}{2}$. Therefore, by using the best performances of robust classifier for a given $\frac{\varepsilon}{2}$ (denoted by $1 - R_{adv}^{\varepsilon/2}$) and computing the $1 - R_{adv}$ for our proposed method (denoted by $1 - R_{adv}^{\varepsilon}$), it is possible to ensure that our presented results are not overclaimed.

According to (Tramer, 2022), for a specific threshold, we have

$$R_{adv}^{\varepsilon} < \text{FPR} + \text{FNR} + R(\text{classifier}), \tag{8}$$

where $R_{adv}^{\varepsilon}$ represent the detection failure of a detection method, FPR and FNR, the False Positive Rate and False Negative Rate at for a specific threshold and $R(\text{classifier})$ the natural error of the considered classifier.

Using the results presented in Tab. 3, we can say that $R_{adv}^{\varepsilon} < 93.27$ for $\varepsilon = 0.125$.

Since $1 - R_{adv}^{\varepsilon/2}$ is around 30 for this specific value , we have that $1 - R_{adv}^{\varepsilon} < 1 - R_{adv}^{\varepsilon/2}$. Therefore, it appears that we do not overclaim our performances which further validate our detector.

