# OpenReview forum: "A Halfspace-Mass Depth-Based Method for Adversarial Attack Detection"
_TMLR — Accepted by TMLR_

### Review · Reviewer_ikB4 · 2022-11-09

**Summary Of Contributions:**

This paper proposes a detector of adversarial examples using halfspace-mass depth, which is a statistical tool that can be used for anomaly detection. The main contribution is in applying this overlooked tool into detection of adversarial examples and also the attempt to design adaptive attacks against the proposed defense.

**Audience:**

Yes

**Broader Impact Concerns:**

Broader impact is sufficiently addressed.

**Claims And Evidence:**

Yes

**Requested Changes:**

The major concern on this work is still on its possible vulnerability to adaptive attacks, although the authors did try evaluating on adaptive attacks. The main issue is, we do not have deep enough understanding on whether depth scores are truly suitable for detecting adversarial examples. The previous LID defense was also believed to be a viable metric before being completely broken by the adaptive attacks. Therefore, I have the following requested changes:

1. discussion on why the half-space mass depth can be a reliable metric for detecting adversarial examples (e.g., inherent power in discriminating adversarial examples).

2. To gain more empirical evidence on the robustness of the detector, I suggest the authors to also evaluate the defense with the following attacks (without clipping the adversarial examples based on certain norm thresholds): 1) targeted CW attacks with different target classes; 2) CW attack with high confidence (controlled by the \kappa parameter).

3. Evaluation on the imagine dataset should be provided. Unlike building robust models, designing highly effective detector (against existing attacks) on imagenet should be feasible.



**Strengths And Weaknesses:**

Strengthes:
1. the proposed method have better performance against existing competitors, in both the attack aware and attack blind settings.
2. the authors also attempted to design adaptive attacks against the proposed detector.

Weaknesses:
1. the detector is a supervised learning algorithm that requires existing attacks to generate adversarial examples for learning proper weight vectors. Therefore, the detector might be vulnerable to future stronger attacks and the (some degree of) vulnerability to adaptive attacks is one possible indicator.
2. the adaptive attacks seem to indicate that, adversaries still can succeed with non-negligible probability (e.g. 30%).
3. the posed norm constraints when generating adversarial examples are not well-justified and other attacks (e.g., targeted CW attack, CW attack with higher confidence).
4. missing evaluation on the imagine dataset.

---

> ### Author Response · Authors · 2023-01-01
> **Reply to reviewer ikB4**
>
> We thank the reviewer ikB4, for their careful reading of the manuscript. We are very **pleased that they recognize that our method performs better in the conscious and blind attack scenario and against the adaptive attacker.**
>
> Below we address the different limitations raised by reviewer ikB4:
> * Reviewer ikB4 asked for more intuition on the halfspace mass depth. The halfspace mass depth is based on a simple idea. It aims to compute an anomaly score by exploring the rank of an element (w.r.t. the training data) on every possible one-dimensional projection. An intermediate score is then assigned to the element according to this rank. Further, the halfspace mass depth value is the expectation of these intermediate (one-dimensional) anomaly scores. It is worth noting that a Monte-Carlo approximation replaces this expectation in practice by computing the average on a finite number of projections. The intuition is that an anomaly should be, on average, farther from the median than normal samples over all possible directions of $\mathbb{R}^d$. _This depth function has the advantage of being simple, interpretable and computationally fast._
>   _Accordingly, in order to address the reviewer comment, we added the previous paragraph to the Appendix B of the revised manuscript._
> * Reviewer ikB4 asked for more attacks. To further assess the strength of our detector, we tested all considered methods on two additional attacks on CIFAR10: a targeted version of $CW_2$ (reported as $CW_{2, targeted}$) and a version of $CW_2$ with higher confidence ($\kappa=30$), both of which don’t clip the adversarial examples according to a specific $L_p$-norm. We reported the following results as AUROC (FPR) in Appendix F:
>
> ||LID|KD-BU|NSS|HAMPER-BA|HAMPER-AA|
> |----------|-------------|----------|-------------|----------|-------------|
> |$CW_{2,\kappa30}$| 54.0 (92.6) | 50.1 (94.8) |  62.0 (88.6) |**96.9 (15.0)** |**100 (0.0)**|
> |$CW_{2,targeted}$| 37.3 (98.2) | 52.6 (93.4) | 46.9 (95.1)| **97.8 (12.6)**| **100 (0.0)**|
>
>
> **These additional attacks further assess the consistent performances of our detector.**
>
> * We thank the reviewer for pointing out the missing dataset. We decided to test all methods on Tiny ImageNet, using a ResNet101, with upscaled images to (224, 224, 3). The results are presented in Tables 5, 6, 7 and 8. Our method still significantly outperforms other methods on this new dataset as it improves over the LID by **5** resp. **12.7** points, over KD-BU by **4.5** points resp. **14.6** and over NSS by **6.8** resp. **16.6** points in terms of AUR resp. FPR.
>
> We hope we have addressed the reviewer's concerns on the experimental evaluation of our detector and that we have improved our evaluation.

---

### Review · Reviewer_NQqi · 2022-12-09

**Summary Of Contributions:**

This paper targets to detect adversarial examples by using a data depth function. Specifically, the author employs halfspace-mass (HM) depth as a metric to detect the adversarial examples that was introduced in anomaly detection. The author provides extensive experiments on both the attack-aware scenario and the blind-attack scenario of the detection. The proposed method, HAMPER, shows better performance than the SOTA model of adversarial detection.

**Audience:**

Yes

**Broader Impact Concerns:**

Author clearly describe the broader impact concerns in the paper.

**Claims And Evidence:**

Yes

**Requested Changes:**

* Request to conduct the test of [1].
> The **last request is a critical** to securing my recommendation whether current detection's claim is correct or not. [1] paper demonstrates that previous adversarial detection may fall into wrong conclusion due to the seemingly easier tasks. [1] propose a sanity check for adversarial detection. Therefore, author should provide the results from the [1].

* Adding relavant and recent baselines
> The rest of the comments are for strengthen the work.


[1] Tramer et al., Detecting Adversarial Examples Is (Nearly) As Hard As Classifying Them \



**Strengths And Weaknesses:**

**Strength** \
This paper is very well-written and easy to follow.
The author provides extensive experiments along with a detailed analysis of the proposed methods.
Furthermore, the proposed methods show much better performance compared to previous works.


**Weakness** \
However, I have a few concerns on this work.
First of all, I think the authors should also provide the detection performance against FGSM, and C&W because it seems those attacks are a more standard benchmark for the adversarial detection task. \
Furthermore, this paper seems to miss some baselines of the adversarial detection.
- Since the current approach is also a motivating technique from anomaly detection, comparing with [1] is also needed. It would show how the current approach is a better technique of anomaly detection to use in the adversarial detection task. Further, based on the previous work [3], [2] seems to show quite good performance on detecting the adversarial examples.
- The author seems to missed recent baselines [3,4].
Lastly, I think author should conduct a test of [1] whether current detection’s claim is correct. Specifically, could the author provide $1-R^\epsilon_{adv-det}$ and $1-R^{\epsilon/2}_{adv}$ values from [1] to compare how much suggested approaches could actually detect the adversarial examples?


[1] Tramer et al., Detecting Adversarial Examples Is (Nearly) As Hard As Classifying Them \
[2] Lee et al., A simple unified framework for detecting out-of-distribution samples and adversarial attacks \
[3] Raghuram et al., A General Framework For Detecting Anomalous Inputs to DNN Classifiers \
[4] Roth et al., The odds are odd: A statistical test for detecting adversarial examples

---

> ### Author Response · Authors · 2023-01-01
> **Answer to Reviewer NQqi**
>
> We would like to first thank reviewer NQqi for their careful reading of the manuscript. We are very glad they acknowledge that our **method shows better results compared to previous works and that they appreciate our efforts to provide a thorough description of the method and explain how it compares to prior approaches.**
>
> We thank reviewer NQqi for pointing out the missing citations. We have added them in the updated version of the manuscript.
>
> Below we address the different limitations raised by reviewer NQqi:
> * Reviewer NQqi asked to further compare our detector using the metric from [1]. $R_{adv}^{\epsilon}$ < FPR + FNR + R(classifier) for a specific threshold. Using the results presented in Table 3, we can say that $R_{adv}^{\epsilon}$ < 93.27 for $\epsilon$ = 0.125. Since $1-R_{adv}^{\epsilon/2}$ is around 30, we have that $1-R_{adv}^{\epsilon} < 1-R_{adv}^{\epsilon/2}$. Therefore, **please notice that we do not overclaim the  performances of our method**. Indeed, these results further validate the value of the proposed detector. In order to further strengthen this in the manuscript, we have added relevant comparison in Appendix G.
> * Reviewer NQqi asked to add more detectors.  We have performed the additional experiments which are now reported in Table 7 and Table 8.  From the results, we observe that our detector outperforms the proposed SOTA methods (i.e. Mahalanobis and Raghuram). We did not run [4] as the method Raghuram outperforms [4] (see [3]). Additionally, due to time constraints, the simulation of the methods did not finish for CIFAR100. We will update them as soon as the simulations are finished.
>
>    **From the current results (see Table 7 and Table 8) our method outperforms the current state of the art.**
>
> * Reviewer NQqi requested evaluation on FGSM and C&W. We did perform those simulations and reported the results in Tables 5 and 6 (see Appendix). Contrary to other works, _we decided to report results for a wider variety of attacks_. Our choice is supported by our experimental results, which show **that the detection methods’ performances widely depend on the considered attack method (e.g., L$_p$ norm, and $\varepsilon$ value)**.
>
> We hope we have addressed the reviewer's concerns on the experimental evaluation of our detector.

---

> > ### Author Response · Authors · 2023-01-05
> > **CIFAR-100 results**
> >
> > We would like to let Reviewer Reviewer NQqi know that we have updated the manuscript with the CIFAR100 results that were missing.

---

### Review · Reviewer_LPTq · 2022-12-18

**Summary Of Contributions:**

The paper proposes to apply the halfspace-mass depth notion in the context of the adversarial detection problem.

The paper introduces HAMPER, a simple supervised method to detect adversarial examples given a trained model. Given an input sample, HAMPER relies on a linear combination of the halfspace-mass depth score. These depth scores are computed w.r.t. a reference distribution corresponding to the training data conditioned per-class and per-layer.

The paper evaluates HAMPER’s performance across popular attack strategies and computer vision benchmark datasets.

**Audience:**

Yes

**Broader Impact Concerns:**

It is OK to investigate the adversarial example detection to enhance the model robustness.

**Claims And Evidence:**

Yes

**Requested Changes:**

As shown in the weakness, it is better to discuss the complexity of the overall algorithm. The time in section 5.4.3 is only the time to train the linear regressor, right? It does not include the time to train the score functions, right? Why not report the overall time and resource consumption? It is better to make the complexity more clear.

The method needs to iterate for each class and each layer. If the complexity is high, it may be hard to use in practice if we switch the dataset or model and repeat the overall high-cost process. It is better to discuss about this.

It seems that the problem formulation and analysis mainly focus on untargeted attacks without considering targeted attack. It looks like to only apply to untargeted attack without further discussions. It is better to provide more discussion about the targeted attack and untargeted attack.

**Strengths And Weaknesses:**

\+ It firstly applies the halfspace-mass depth notion in the context of the adversarial detection problem.

\-  The technical contribution may be limited. The paper mainly adopts the halfspace-mass depth method following previous papers such as Chen et al. 2015 which originally use this method for anomaly detection. The novelty may not be significant. The main algorithms and hyper-parameters mainly follow Chen et al. 2015. The aggregation procedure follow several supervised settings to simply use a linear regressor in a supervised manner. The technical contribution may be limited.

\- The complexity can be high. The score functions need to iterate over each class in the dataset and each layer in the model. The complexity may be high if the number of classes and layers are high for large datasets or models. It is better discuss the overall complexity. The authors show the training time and testing time. But this time is only the time to train the detectors with a linear regressor. The time to compute the halfspace-mass depth score function is not reported. Without the score functions, simply use a linear regressor does not work.

\- It can be hard to use the method in practice. The score functions need to be trained for each class in the dataset and each layer in the model. If the dataset is switched or we use another model, we need to repeat the whole process. If the whole process takes a lot of time and efforts with high complexity. It may be not practical.

\- The problem formulation mainly uses untargeted attack without a target label. So it is ok as long as the classified label differs from the true label. But for the attacks with targeted attack version, we can assign a target label (which differs from the true label) and the classified label should be the target label. The analysis in Section 4 also depends on untargeted attack to show the average number of adversarial examples per class. In targeted attack, as we can control the target label, the distribution per class can be very different.  It seems that the problem formulation and analysis mainly focus on untargeted attacks without considering targeted attack. It is better to provide more discussion.

---

> ### Author Response · Authors · 2023-01-01
> **Answer to Reviewer LPTq**
>
> We thank reviewer LPTq for their careful reading of the manuscript. We are glad they **find the use of halfspace mass depth in the scope of adversarial attack detection interesting**.
>
> Below we address the different limitations raised by reviewer LPTq:
> * LPTq claims that our work suffers from limited technical novelty.
>   * The halfspace-mass depth **_has been both overlooked and never used_** by the deep learning community and, more specifically, in the Computer Vision community.
>   * Our experiments demonstrate the superiority of our method over previously introduced detectors (by up to 26.6 \%), which justifies its relevance for submission to TMLR.
>   * We agree that halfspace-mass depth is borrowed from statistical literature, but _we argue that this is the case for most previously published papers in this field_. For example, [2] (see Reviewer NQqi) uses the Mahalanobis distance borrowed from [a], [5] uses the Fisher-Rao distance borrowed from [b],  LID uses the Local Intrinsic Dimensionality, that was already know in 1989 [c]. Additionally, **our detector involves several steps** (e.g., class-conditioning, score normalization), the halfspace-mass depth only represents one of them.
>   * Last, our method brings novel ideas to the Computer Vision community regarding defenses. Indeed, if the statistical community knows the halfspace-mass depth, it has never been used for deep learning applications. We believe it is a very well-suited tool for detecting attacks due to its efficiency and non-differentiability.
> **Our detector, which outperforms existing detectors by a significant margin, shows that the halfspace-mass depth can be used with a deep neural network. We believe that these results open up new avenues of research and may provide ideas for researchers to further investigate those tools from statistics.**
> * Reviewer LPTq has concerns about the time complexity of the method. In Table 4, we reported the training times of the different detectors' and inference times on the dataset. We thank reviewer LPTq for pointing this out. We added a clarification in the caption. In addition, we would like **to point out that each depth is computed independently, which allows for efficient computation through parallelization.**
> * Reviewer LPTq claims that we did not evaluate our method on targeted attacks. In the manuscript, we considered HOP, which is indeed a targeted attack. We refer the reviewer to Appendix C (see Tables 5 and 6) for our results on HOP: HAMPER outperforms LID by up **31.3 points**, KD-BU by up to **28.2 points**, NSS by up to **34.6 points** and [3] by up to **4.5 points**, and [2]  by up. to **23.3 points**  in terms of AUROC. In addition, as requested by Reviewer ikB4, we performed an additional simulation using the targeted version of the CW attack, and a more confident version of the CW attack. We refer the reviewer to our answer to reviewer ikB4 for the numerical values.  **Overall, our method outperforms existing detectors both on targeted and untargeted attacks.**
>
> _We hope we have addressed the reviewer's questions on the novelty of our method, the time complexity and the comparison with targeted attacks._
>
> **References:**
>
> [a] Mahalanobis, P. C. (1936). On the generalized distance in statistics. National Institute of Science of India.
>
> [b] Rao, C. R. (1992). Information and the accuracy attainable in the estimation of statistical parameters. In Breakthroughs in statistics(pp. 235-247). Springer, New York, NY.
>
> [c] Passamante, A., Hediger, T., & Gollub, M. (1989). Fractal dimension and local intrinsic dimension. Physical Review A, 39(7), 3640.
>
> [5] Gomes, E. D. C., Alberge, F., Duhamel, P., & Piantanida, P. (2022). Igeood: An Information Geometry Approach to Out-of-Distribution Detection. arXiv preprint arXiv:2203.07798.
>
> [2] Lee et al. (2018). A simple unified framework for detecting out-of-distribution samples and adversarial attacks. 32nd Conference on Neural Information Processing Systems (NeurIPS 2018).

---

### Decision · Action_Editors · 2023-01-23

**Recommendation:** Accept with minor revision

**Comment:**

TMLR's criteria are "claims supported by evidence" and "interest to audience". I believe the results and analysis in this work is indeed interesting but the claims need to be reduced. I would be willing to consider a revision that addresses the concerns laid out above (namely around computational efficiency and adaptive attacks). The revision should (1) reduce claims about computation efficiency to be more in line with reviewer LPTq's comments, (2) provide more explanation or evaluations about the relation of depth scores to detecting adaptive adversarial examples, as suggested by ikB4, and (3) explain why the adaptive attacks are black-box (which weakens the adversary) rather than white-box.

**Audience:**

While the authors did a good job of addressing attack-aware and blind-attack scenarios, ultimately the important question on the security community is how well the defense does against adaptive attacks. The ICML 2018 best paper [1] went to great ends by attacking more than 10 defenses to establish the fact that most defenses in deep learning (especially to cifar scale image classifiers such as those considered in this paper) are brittle against adaptive attacks. As Reviewer ikB4 rightly pointed out, "The major concern on this work is still on its possible vulnerability to adaptive attacks, although the authors did try evaluating on adaptive attacks. The main issue is, we do not have deep enough understanding on whether depth scores are truly suitable for detecting adversarial examples." This was not very well addressed by the authors. I believe any defense/detection paper will be lacking if it doesn't have a clear avenue of defense against adaptive attacks.

[1] Athalye, Anish, Nicholas Carlini, and David Wagner. "Obfuscated gradients give a false sense of security: Circumventing defenses to adversarial examples. ICML." arXiv preprint arXiv:1802.00420 (2018).

**Claims And Evidence:**

This paper proposes using the linear combination of the halfspace-mass depth score, a tool from statistics for anomaly detection, to detect adversarial examples.

Reviewers NQqi and ikB4 asked for testing the proposed detection method on additional attacks and image datasets, to which the authors promptly provided to the satisfaction of the reviewers.

Reviewer LPTq had concerns about the computational complexity being high, to which the authors responded by saying that the algorithm for computing depth can be parallelized. However, as LPTq points out, certain optimizations can be done for baselines too. To have a clearer picture on the complexity, a more nuanced and controlled comparison is needed. As such, I don't believe the claim that "the halfspace-mass (HM) depth exhibits attractive properties such as computational efficiency" as written in the abstract is backed up enough by evidence and exposition.

---

> ### Author Response · Authors · 2023-03-15
> **Answer to Action Editor**
>
> We would like to first thank the Action Editor for their careful reading of the manuscript, and answers we made to the reviewers. We are very glad they acknowledge that our method shows better results compared to previous works and that they appreciate our efforts to provide additional content to the reviewers.
>
> Below we address the comments made by the Action Editor:
>
> * We reduced our claim about computation efficiency in the abstract.
>  * We provided more explanation about the relation of depth scores to detecting adaptive adversarial examples, as suggested by ikB4, and the reason for using blackbox adaptive attacks. We add the following clarification in Sec. 5.3. “The importance of attacking defenses with adaptive attacks has increased recently (Abusnaina et al., 2021; Raghuram et al., 2021). As mentioned in Sec. 2.3, the Backward-Pass Differentiable Attack (BPDA; Athalye et al., 2018a) are based on the possibility to find a suitable surrogate to the non-differentiable parts of any defense. However, deriving a suitable differentiable surrogate of the halfspace-mass depth remains an open research question which has never been tackled. As a matter of fact, the only attempts to approximate a non-differentiable depth was performed on the Tuckey depth in She et al. (2021), with very poor results (Dyckerhoff et al., 2021). It is worth to point out that, if approximating the Tuckey depth is already hard, finding a differentiable surrogate for IRW would be even harder because it involves several non-differentiable components (the indicator function, the function x $\rightarrow$ min{x, 1−x} and the need to compute a ranking). Finding a differentiable suitable surrogate to attack $\texttt{HAMPER}$ would, therefore, require a substantial effort and should be rigorously handled. As a consequence, we have to rely on adaptive blackbox attackers, as suggested in Tramer et al. (2020), to attack $\texttt{HAMPER}$." We also add comments in the conclusion.
>
> *We hope we have addressed the Action Editor's comment*

---

> > ### Comment · Action_Editors · 2023-03-15
> > **looks good to me**
> >
> > thank you for the efforts!